# Revisiting Data Augmentation for Ultrasound Images

**Adam Tupper**                                                                    *adam.tupper.1@ulaval.ca*
*Institut Intelligence et Données (IID), Université Laval, Mila*

**Christian Gagné**                                                              *christian.gagne@gel.ulaval.ca*
*Institut Intelligence et Données (IID), Université Laval*
*Canada-CIFAR AI Chair, Mila*

**Reviewed on OpenReview:** *https: // openreview. net/ forum? id= iGcxlTLIL5*

## Abstract

Data augmentation is a widely used and effective technique to improve the generalization performance of deep neural networks. Yet, despite often facing limited data availability when working with medical images, it is frequently underutilized. This appears to come from a gap in our collective understanding of the efficacy of different augmentation techniques across different tasks and modalities. One modality where this is especially true is ultrasound imaging. This work addresses this gap by analyzing the effectiveness of different augmentation techniques at improving model performance across a wide range of ultrasound image analysis tasks. To achieve this, we introduce a new standardized benchmark of 14 ultrasound image classification and semantic segmentation tasks from 10 different sources and covering 11 body regions. Our results demonstrate that many of the augmentations commonly used for tasks on natural images are also effective on ultrasound images, even more so than augmentations developed specifically for ultrasound images in some cases. We also show that diverse augmentation using TrivialAugment, which is widely used for natural images, is also effective for ultrasound images. Moreover, our proposed methodology represents a structured approach for assessing various data augmentations that can be applied to other contexts and modalities.

## 1 Introduction

Data augmentation is an essential component of deep learning. It not only improves generalization, but it is also a core component of many self- and semi-supervised learning algorithms. However, while data augmentation is ubiquitous for training deep neural networks on natural images (i.e., images of human-scale scenes captured by ordinary digital cameras), when it comes to training such models on medical images its proper usage is not as common and clearly understood (Chlap et al., 2021; Garcea et al., 2023). This is despite the difficulties we face collecting sufficient data due to privacy protections and high acquisition and annotation costs.

The under-utilization of augmentation when working with medical images suggests a weaker understanding of the effectiveness of different operations and strategies. Often, we simply apply photometric and geometric transformations proposed from natural images as is, without rigorous testing. However, the low uptake indicates that findings from natural images may not translate well to medical images. This is not surprising given that the size of the objects of interest and the relevance of specific textures may differ significantly for doing detection, classification or segmentation tasks from natural images compared to other modalities such as microscopy, X-ray, and ultrasound, to name a few.

A lack of comparative studies featuring controlled experiments that evaluate various techniques over different tasks, datasets, and imaging modalities has created a gap in our understanding of data augmentation for medical images. While there are several excellent literature surveys on this topic (Chlap et al., 2021; Garcea et al., 2023), relying solely on surveys leaves us at risk of falling foul of publication bias (i.e., the file-drawer

effect). In addition, these surveys highlight the difficulty in drawing conclusions on which transforms are most effective, since there are many confounding variables. Ultimately, drawing conclusions from literature surveys alone is not enough. This problem needs to be addressed more rigorously and systematically through an experimental approach. In this work, we evaluate the effectiveness of data augmentation techniques for deep neural networks in ultrasound image analysis.[1] Our investigation reveals several key findings that provide practical guidance for implementing data augmentation in ultrasound image analysis:

1. Traditional domain-independent augmentations are effective, even more so in many cases than ultrasound-specific augmentations. They can be leveraged to achieve quick performance improvements before investing time and resources in developing custom techniques.

2. The impact of individual augmentations varies substantially across both domains (cardiac vs. liver ultrasound) and tasks (classification vs. semantic segmentation), with notable differences even between similar tasks on the same dataset.

3. While these variations might suggest the need for careful task-specific tuning of augmentation strategies, we find that applying a diverse set of augmentations using the simple TrivialAugment strategy (Müller & Hutter, 2021) achieves substantial performance gains with limited tuning of the augmentation set.

The remainder of this paper is organized as follows. First, we discuss the prevalence of data augmentation for ultrasound image analysis using deep learning, ultrasound-specific augmentations, and previous studies of data augmentations. Second, we present our benchmark that serves as the foundation for our analyses. Third, we provide an in-depth description of the ultrasound-specific augmentations included in our study. Fourth, we present our analyses of individual augmentations and TrivialAugment. Finally, we discuss the implications of our results.

## 2  Background

As previously mentioned, the use of data augmentation for ultrasound imaging is far less common than for natural image analysis. We start by presenting concrete analysis supporting this observation, then discuss proposals for ultrasound-specific data augmentations, and finally examine prior studies comparing the efficacy of different data augmentations for medical image analysis.

### 2.1  Data Augmentation in Ultrasound Image Analysis

To understand data augmentation practices for ultrasound image analysis with deep learning, we analyzed the use of data augmentation on 10 different publicly available ultrasound image datasets (Xu et al., 2023; Butterfly Network, 2018; Leclerc et al., 2019; Byra et al., 2018; Basu et al., 2022; Zhao et al., 2023; Singla et al., 2023; Born et al., 2021; Chen et al., 2024; Stanford AIMI Center, 2021) covering 11 regions of the body. We describe these datasets comprehensively in the following section as they form the basis of our benchmark. For now, we focus on the snapshot they provide of the use of data augmentation in ultrasound imaging.

Of the 557 citations of the datasets catalogued by the Clarivate Web of Science platform[2] as of August 2024, we identified 165 studies that used these datasets to train deep neural network models for classification and segmentation tasks. Among these studies, more than half (85 of 165) used no data augmentation at all when training their models. Out of those remaining, 48 used three or less augmentations and only only 13 used six or more. This pales in comparison to the large sets of 14 augmentations used in common data augmentation strategies such as AutoAugment (Cubuk et al., 2019), RandAugment (Cubuk et al., 2020), and TrivialAugment (Müller & Hutter, 2021).

---

[1]Our code, documentation and benchmark are available at `https://github.com/adamtupper/ultrasound-augmentation`.
[2]`www.webofscience.com`

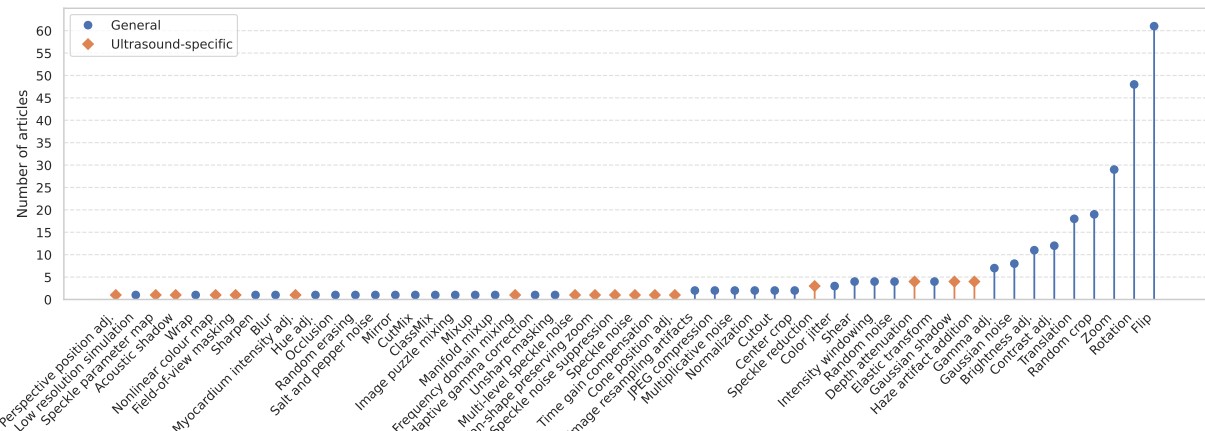

Figure 1: The popularity of different augmentation techniques among 165 studies from the ultrasound literature, showing moderate adoption of common methods but limited adoption of ultrasound-specific methods.

Upon examining the popularity of different augmentations, the list is dominated by natural image augmentations commonly found in deep learning frameworks. As presented in Fig. 1, the most popular augmentations are classic geometric transforms that are known to perform well for natural image tasks, such as image flipping, rotation, zoom/scaling, random cropping, and translation. However, even among these most popular techniques, there is a steep decline in their use. The first augmentations designed specifically for ultrasound images, that is Gaussian shadowing (Smistad et al., 2018), haze artifact addition, depth attenuation and speckle reduction (Ostvik et al., 2021), are used in only three or four articles. In fact, these are the only ultrasound-specific augmentations used in multiple studies in our sample. We discuss these, along with other ultrasound-specific augmentations, in the following section.

Another interesting observation is the lack of adoption of "modern" data augmentation strategies (e.g., RandAugment, TrivialAugment, etc.) among these studies, suggesting skepticism surrounding their efficacy from researchers working on medical image analysis using deep learning. Instead, a common pattern we found is that researchers tend to focus on simple, hand-crafted, fixed sequences of augmentations that reflect plausible differences that might arise in real-world settings. This is despite the fact that these stronger, "unrealistic" strategies have proved effective in other modalities. The strategies adopted in deep learning for medical image analysis are reminiscent of the strategies used a decade or more ago for general computer vision. This gap may reflect that ultrasound imaging simply lags behind natural image processing in adopting more recent techniques. Our work demonstrates the effectiveness of one such modern augmentation strategy for ultrasound image analysis, aiming to provide evidence that encourages wider adoption of these methods in the field.

## 2.2 Ultrasound-Specific Data Augmentation

While domain-independent augmentations are the most frequently used, a variety of ultrasound-specific techniques have also been proposed. These target unique characteristics of ultrasound images to better simulate different machines and imaging conditions.

A common focus is noise manipulation. Techniques such as Multi-Level Speckle Noise (Monkam et al., 2023), Speckle Distortion (Ramakers et al., 2024), and Speckle Noise (Wang et al., 2022) all try to simulate realistic speckle noise, while Speckle Noise Suppression (Monkam et al., 2023) and Speckle Reduction (Ostvik et al., 2021) aim to reduce it. Additional methods, including Haze Artifact (Ostvik et al., 2021) and multiplicative noise, simulate other realistic types of noise. Other methods simulate occlusions or variations in the field of view, similar to Cutout (DeVries & Taylor, 2017) and random cropping. Some darken regions to mimic acoustic shadows (Smistad et al., 2018; Singla et al., 2022; Ramakers et al., 2024), while Fan-Shape Preserv-

ing Zoom (Singla et al., 2022), Field-of-View Masking (Pasdeloup et al., 2023), and Fan-Preserving Crop (Ramakers et al., 2024) all reduce the field of view.

To account for variability in probe orientation, Cone Position Adjustment and Perspective Position Adjustment (Sfakianakis et al., 2023) simulate changes in angle and rotation. Other methods adjust image intensity and contrast. These include Myocardium Intensity Adjustment (Sfakianakis et al., 2023), which enhances specific cardiac regions, Tirindelli et al.'s (2021) signal-to-noise ratio augmentation, which modifies the relative intensity of spinal structures, and Non-linear colour mapping (Pasdeloup et al., 2023). Similarly, Singla et al. (2022) transform images into "speckle parameter maps" to emphasize different tissue structures.

Some augmentations are variations of general image mixing methods, like mixup (Zhang et al., 2018) and CutMix (Yun et al., 2019). Frequency-Domain Mixing (Ding & Han, 2024) mixes images in the frequency domains, while Mixed-Example (Lee et al., 2021) mixes images according to specific patterns. Others replicate ultrasound-specific artifacts like depth attenuation (Ostvik et al., 2021; Singla et al., 2022), reverberation (Ramakers et al., 2024; Tirindelli et al., 2021), or deformation (Tirindelli et al., 2021; Ramakers et al., 2024).

Despite their potential, many of these augmentations face barriers to adoption. Some are domain-specific or require precise segmentation maps, while others lack public implementations or have only been evaluated on a narrow set of tasks. Although our focus is on data augmentation, it would be amiss to ignore the large body of research on synthetic data generation using generative models (Kebaili et al., 2023). This offers a complementary approach, though it introduces its own challenges, such as generating accompanying segmentation masks or focusing on spurious correlations (e.g., correlation with medical devices) rather than relevant features. These issues aside, augmentation and synthetic data could significantly enhance model robustness and generalization when used together.

### 2.3 Previous Studies

Despite the proven benefits of data augmentation, there are few comparative studies evaluating which techniques and strategies are most effective for medical images, despite the fragmented usage patterns and extensive literature on specialized methods. This has led to calls for more such studies (Garcea et al., 2023). Our study extends these previous studies in several notable ways. First, only a single previous study included ultrasound images (Rainio & Klén, 2024) and was limited to a single task. Furthermore, in this case and other studies on different imaging modalities (Bali & Mahara, 2023; Castro et al., 2018; Haekal et al., 2021; Hussain et al., 2018; Rama et al., 2019), the effectiveness of each augmentation was only tested when used offline – that is, used once before training to increase the size of the training set. In the case of Lo et al. (2021), augmentation policies were learned via a policy learning algorithm, but the effectiveness of individual augmentations within the defined search space was not assessed. The same goes for Liu et al. (2023) who proposed an alternative augmentation strategy to TrivialAugment. Finally, Eaton-Rosen et al. (2018) compared the effectiveness their own sampling mixing augmentation against mixup (Zhang et al., 2018), which is frequently used in natural image settings.

## 3 UltraBench

One of the limitations of previous studies is a lack of comparisons across multiple domains. This challenge arises from the absence of ultrasound image analysis tasks in existing medical image analysis benchmarks, such as MedMNIST (Yang et al., 2023), MedSegBench (Kuş & Aydin, 2024) and the Medical Segmentation Decathlon (Antonelli et al., 2022). To assess different augmentations for ultrasound image analysis, we created a benchmark of ultrasound image analysis tasks. The benchmark includes 14 tasks (7 classification, 7 segmentation) from 10 public datasets of 2D "fan-shape" ultrasound images captured with either convex and phased array ultrasound probes. These were used in the previous analysis of data augmentation usage and cover 11 regions of the body.

The following subsections outline the tasks defined within each dataset. Except for cases where data splits are predefined by the dataset's original authors, we split each dataset into training, validation, and test images using a 7:1:2 split, using patient identifiers where applicable to ensure that there is no patient overlap

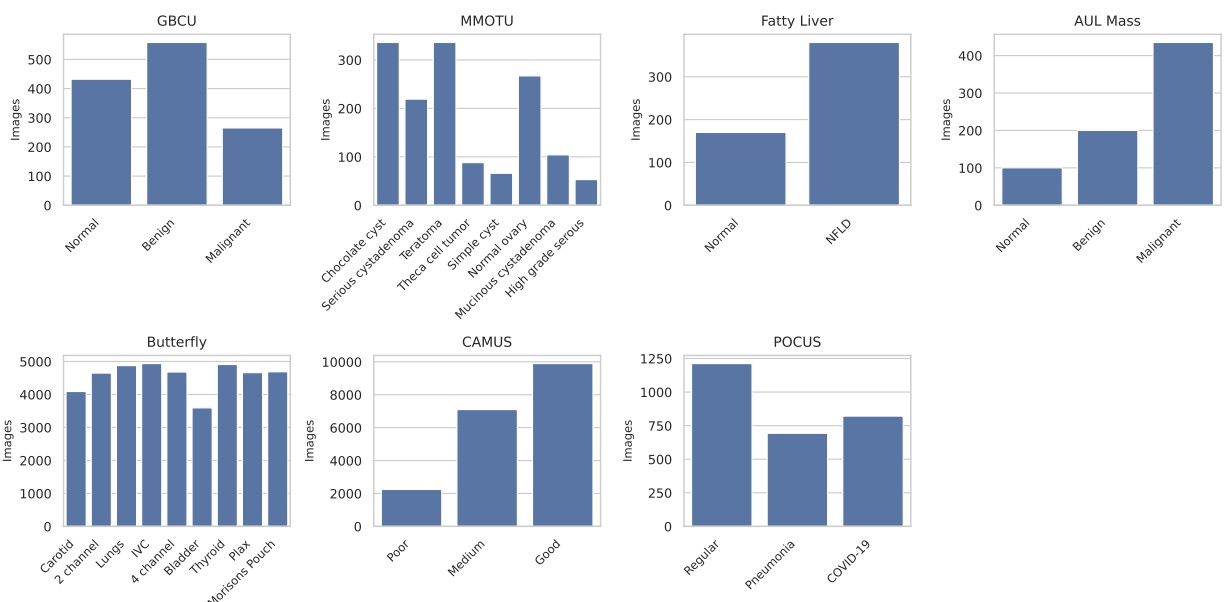

Figure 2: The class distributions for each image classification task included in UltraBench.

between the sets. Fig. 2 shows the class distribution for each classification task, while Fig. 3 provides examples of the images and segmentation masks for each segmentation task.

We also created approximate scan segmentation masks for each dataset using morphological operations to support the ultrasound-specific augmentations described in the following section. Examples of the masks generated for each dataset are provided in Appendix A.

**Annotated Ultrasound Liver** The Annotated Ultrasound Liver (AUL) dataset (Xu et al., 2023) consists of 735 images, including 435 with malignant masses, 200 with benign masses, and 100 with no masses. Each image is of a different patient, with a mean width of 945.33 px ($\sigma$: 142.46 px, min: 440 px, max: 1388 px) and height of 713.80 px ($\sigma$: 94.81 px, min: 341 px, max: 910 px). With the exception of one image that is missing the outline of the liver, each image is annotated with the outline of the liver and the outline of the masses (if present). In addition, each image is labeled *malignant*, *benign*, or *normal* according to the presence of malignant, benign, or no masses in the image, respectively.

We define two segmentation tasks and one classification task on the AUL dataset: a liver segmentation task using the 734 images with liver annotations, a mass segmentation task on all 735 images, and a mass classification task to classify the images according to the type of mass present in the images.

**Butterfly** The Butterfly dataset (Butterfly Network, 2018) was released for the 2018 MIT Grand Hack. It consists of ultrasound images of multiple body regions acquired using the Butterfly iQ point-of-care ultrasound device from 31 patients. The images are divided into nine groups according to the organ being imaged (*morison's pouch*, *bladder*, *heart (PLAX view)*, *heart (4-chamber view)*, *heart (2-chamber view)*, *IVC*, *carotid artery*, *lungs*, and *thyroid*). We use these labels to define a nine-class image classification task. In total, the dataset consists of 41,076 images, 34,325 of which are allocated to training and validation, while 6751 are reserved for testing. The images have an average width of 415.57 px ($\sigma$: 31.16 px, min: 360 px, max: 462 px) and height of 500.80 px ($\sigma$: 36.16 px, min: 384 px, max: 512 px). We split the training and validation images using an 80:20 split into training and validation sets, ensuring that there is no patient overlap between the sets.

**CAMUS** The Cardiac Acquisitions for Multi-structure Ultrasound Segmentation (CAMUS) dataset (Leclerc et al., 2019) consists of apical four-chamber and two-chamber view cardiac ultrasound sequences

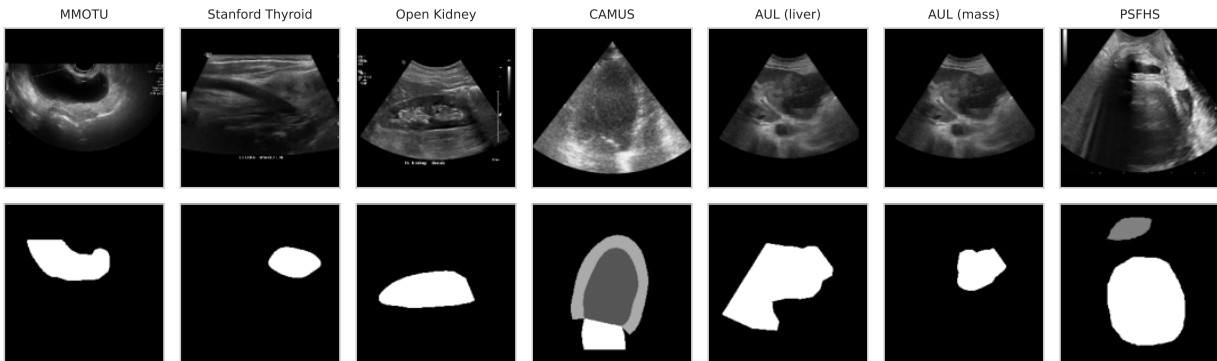

Figure 3: An example image and segmentation mask for each segmentation task included in UltraBench.

from 500 patients for a total of 19,232 images. The metadata provided with each image includes segmentation masks for the left ventricle endocardium, myocardium, and the left atrium. Each image is also labeled according to the quality of the scan (*poor*, *medium*, and *good*). The images have an average width of 597.58 px ($\sigma$: 102.80 px, min: 323 px, max: 1181 px) and an average height of 491.58 px ($\sigma$: 77.83 px, min: 292 px, max: 973 px). For the CAMUS dataset, we include two tasks: image quality classification, as was explored by Nazar et al. (2024), and cardiac structure segmentation, the original segmentation task.

**Fatty Liver**  The Dataset of B-mode fatty liver ultrasound images (Byra et al., 2018), referred to simply as the Fatty Liver dataset from here on, contains 550 liver ultrasound images from 55 patients, with 38 suffering from non-alcoholic fatty liver disease (NFLD; defined as >5 % of hepatocytes having fatty infiltration). The images all have a resolution of $436 \times 636$ pixels. The associated binary classification task is to classify the images into *normal* and *NFLD*.

**GBCU**  The Gallbladder Cancer Ultrasound (GBCU) dataset (Basu et al., 2022) contains a total of 1255 annotated abdominal ultrasound images (consisting of 432 *normal*, 558 *benign*, and 265 *malignant* images) collected from 218 patients (71 normal, 100 benign, and 47 malignant). The images have an average width of 1204.95 px ($\sigma$: 85.43 px, min: 854 px, max: 1156 px) and an average height of 854.64 px ($\sigma$: 36.11 px, min: 688 px, max: 947 px). The dataset is already split into training and testing sets containing 1133 and 122 images, respectively. We further split the training set into training and validation sets using an 90:10 split. While there is no patient overlap between the training and test sets, we cannot guarantee that there is no patient overlap between the training and validation sets since all patient information was removed before the dataset was published. The associated task is to classify the images according to the three classes.

**MMOTU**  The Multi-Modality Ovarian Tumor Ultrasound (MMOTU) dataset (Zhao et al., 2023) is an ovarian cancer dataset consisting of 2D ultrasound and contrast-enhanced ultrasonography images. In this case, we are interested only in the ultrasound images. In total, there are 1469 2D ultrasound images, each accompanied by semantic segmentation masks that identify the tumor in the image. In addition, each image is labeled according to the presence of each type of tumour (*chocolate cyst*, *serous cystadenoma*, *teratoma*, *theca cell tumour*, *simple cyst*, *normal ovary*, *mucinous cystadenoma*, and *high grade serous cystadenocarcinoma*). This allows us to define two tasks on this dataset: binary tumour segmentation and multi-class tumour type classification. As is the case of the GBCU dataset, this dataset is pre-split into training and testing sets, with 1000 examples collected from 171 patients in the training set and 469 examples collected from 76 patients in the test set. We further split the training set into training and validation sets using an 80:20 split. However, as all patient information has been removed from the dataset, we cannot guarantee the absence of patient overlap between the training and validation subsets. The images have an average width of 550.84 px ($\sigma$: 55.38 px, min: 266 px, max: 794 px) and an average height of 762.04 px ($\sigma$: 238.06 px, min: 302 px, max: 1135 px).

**Open Kidney** The Open Kidney Ultrasound dataset (Singla et al., 2023) consists of 514 B-mode kidney ultrasound images, each from a distinct patient. The images are annotated with kidney capsule pixel masks, which permits two separate semantic segmentation tasks: kidney capsule and a more fine-grain kidney regions segmentation. However, the limited amount of data relative to the complexity of the region segmentation task means that training informative, effective models in this setting is not possible. The images have an average width of 1061.92 px ($\sigma$: 200.44 px, min: 640 px, max: 1920 px) and an average height of 773.71 px ($\sigma$: 94.88 px, min: 480 px, max: 1080 px). To minimize distribution drift between the splits, we stratify by view (*transverse*, *longitudinal*, and *other*) when creating the training, validation, and test splits.

**POCUS** The Point-of-care Ultrasound (POCUS) dataset (Born et al., 2021) is a collection of convex and linear probe lung ultrasound images and videos from different sources that was created for the diagnosis of COVID-19. We use the 142 convex probe videos and 29 convex probe images distributed by the authors and follow the procedure described in their original paper to process them, sampling the videos at a rate of 3 Hz, up to a maximum of 30 frames, and grouping the frames by video to prevent data leakage between the train, validation, and test splits. In total, we extract 2726 examples. Each image is labeled by the pathology (*healthy*, *pneumonia*, *covid*) yielding a three-class classification problem. The images have an average width of 499.22 px ($\sigma$: 205.39 px, min: 139 px, max: 1280 px) and an average height of 462.84 px ($\sigma$: 167.07 px, min: 139 px, max: 1080 px).

**PSFHS** The PSFHS dataset (Chen et al., 2024) is a dataset for fetal head and pubic symphysis segmentation, comprising 1358 images from 1124 patients. Each image is accompanied by pixel-level segmentation masks for the fetal head and pubic symphysis, supporting a three-class image segmentation task (*background*, *pubic symphysis*, and *fetal head*). The images all have a resolution of $256 \times 256$ px.

**Stanford Thyroid** The Stanford Thyroid Ultrasound Cine-clip dataset (Stanford AIMI Center, 2021), referred to simply as the Stanford Thyroid dataset from here on, is a dataset of 192 thyroid nodule ultrasound cine-clips (videos) collected from 167 patients. The images in each sequence are associated with pixel-level nodule segmentation masks, patient demographics, lesion size and location, TI-RADS descriptors, and histopathological diagnoses. We use the nodule masks for a thyroid nodule segmentation task. In total, there are 17,412 images all with a resolution of $1054 \times 802$ px.

## 4 Ultrasound Image Augmentations

Of the ultrasound-specific augmentations presented in Fig. 1, very few have been used beyond the original works. The exceptions are the depth attenuation, haze artifact, and speckle reduction augmentations proposed by Ostvik et al. (2021), and the Gaussian shadow augmentation proposed by Smistad et al. (2018). However, their usage still pales in comparison to the most popular augmentations. These augmentations may see limited use not only due to their absence from standard libraries (e.g., Torchvision, MONAI, Albumentations) and lack of open-source implementations, but also because their effectiveness has not been widely demonstrated. To address these barriers, we provide implementations compatible with these libraries based on the original articles' descriptions, which are detailed below. The implementations are provided as a Python package alongside our source code.

### 4.1 Depth Attenuation

The depth attenuation augmentation proposed by Ostvik et al. (2021) is designed to mimic the loss of energy of the ultrasound wave energy as it moves through the body, which results in a gradual drop in intensity with distance from the probe. In Ostvik et al. (2021), this is implemented as applying a "varying degree of intensity attenuation along the radial direction". Guided by the visualizations of the attenuation maps in their paper, and knowing that the intensity of the wave should decrease exponentially with distance, we implement the augmentation as follows.

Assuming the ultrasound fan is oriented such that the probe is positioned at the middle-top of the image, we create an attenuation map that is used to scale the intensity of each pixel of the ultrasound scan mask

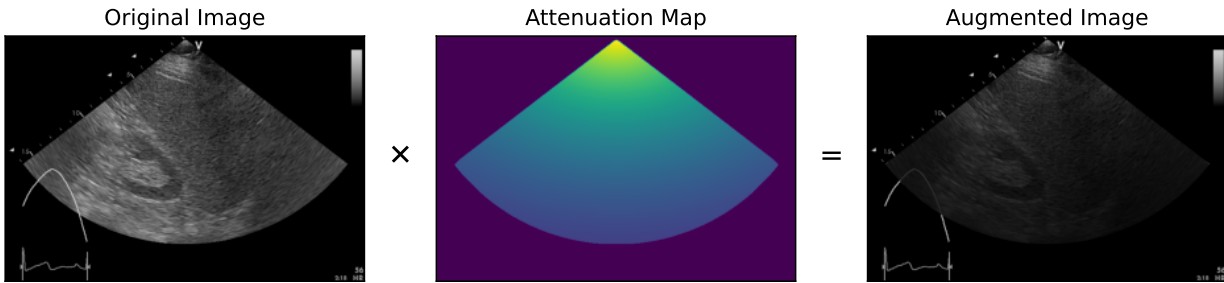

Figure 4: An example of the depth attenuation augmentation on the AUL dataset with a maximum attenuation ($\lambda$) of 0 and attenuation rate ($\mu$) of 1.5.

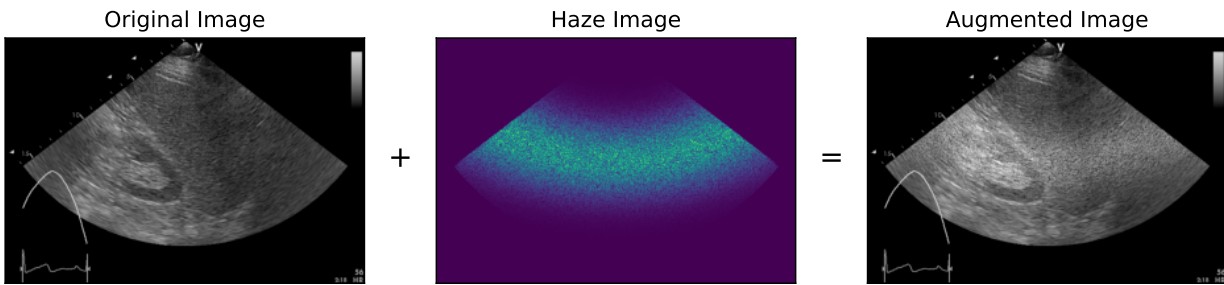

Figure 5: An example of the haze artifact augmentation on the AUL dataset with a radius $r = 0.5$ and $\sigma = 0.1$.

$S$ in the original image $I$, as illustrated in Fig. 4. The resulting image $I'$ is given by

$$I'(x, y) = A(x, y) \odot S(x, y) \odot I(x, y). \tag{1}$$

The attenuation map $A$ is calculated as

$$A(x, y) = (1 - \lambda) \exp(-\mu d) + \lambda, \tag{2}$$

where $d = \sqrt{(x - 0.5)^2 + y^2}$. The maximum attenuation $\lambda$ and attenuation rate $\mu$ are configurable parameters. By default, $\lambda$ is set to 0 and to generate variation $\mu$ is sampled uniformly from the range $[0, 3)$.

## 4.2 Haze Artifact Addition

Acoustic haze is a semi-static noise band that is sometimes present in ultrasound images. To mimic this, Ostvik et al. (2021) proposed a haze artifact augmentation that applies static with a Gaussian profile at a fixed distance (radius) from the probe. Guided by their illustrations, we implement this augmentation by generating a haze image $H$ that is added to the pixels that lie within the ultrasound scan mask $S$ in the original image $I$.

For a given haze radius $r$ and standard deviation $\sigma$ that controls the spread of the noise, the haze image $H$ is calculated as

$$H(x, y) = \frac{1}{2} u \exp(-\frac{(d - r)^2}{2\sigma^2}), \tag{3}$$

where $d = \sqrt{(x - 0.5)^2 + y^2}$ and $u \sim U(0, 1)$. This results in an image similar to that shown in Fig. 5. By default, $r \sim U(0.05, 0.95)$ and $\sigma \sim U(0, 0.1)$.

## 4.3 Gaussian Shadow

To mimic acoustic shadows that occur due to air or tissue blocking acoustic waves from penetrating deeper, the Gaussian shadow augmentation proposed by Smistad et al. (2018) generates and applies two-dimensional

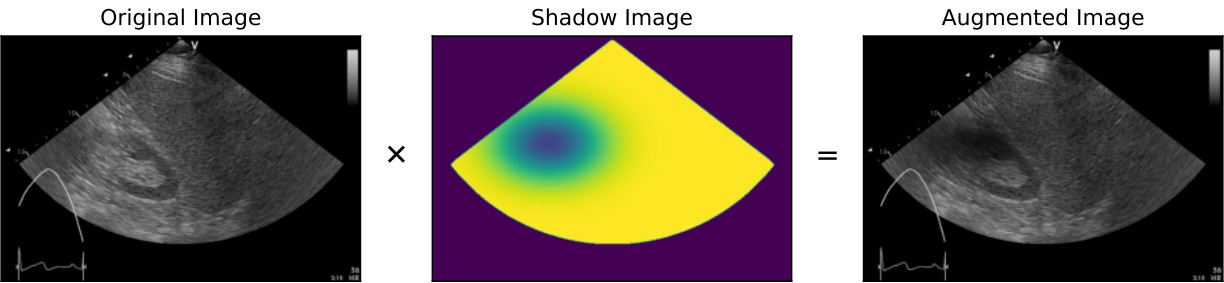

Figure 6: An example of the Gaussian shadow augmentation on the AUL dataset with strength $s = 0.8$, and $\sigma_x = \sigma_y = 0.11$.

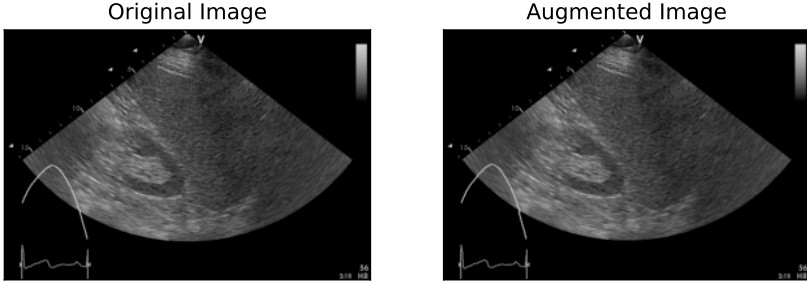

Figure 7: The speckle reduction augmentation applied to an image from the AUL dataset with $\sigma_{\text{spatial}} = 1.0$ and $\sigma_{\text{color}} = 1.0$.

Gaussian shadows with randomly selected parameters. The shadow centre $(\mu_x, \mu_y)$ is randomly positioned in the image, while its dimensions $(\sigma_x, \sigma_y)$ are sampled uniformly between 0.1 and 0.4 of the image size. This upper limit is lower than the 0.9 used by (Smistad et al., 2018), who originally designed the augmentation for rectangular linear probe images, rather than fan-shaped convex probe images. The shadow strength $s$ is sampled uniformly between 0.25 and 0.8. The Gaussian shadow image $G$ is then calculated as

$$G(x,y) = 1 - s \exp\left(-\frac{(x-\mu_x)^2}{2\sigma_x^2} - \frac{(y-\mu_y)^2}{2\sigma_y^2}\right). \tag{4}$$

Finally, the augmented image $I'$ is generated by the pixel-wise multiplication of $G$, the ultrasound scan mask $S$, and the original image $I$:

$$I'(x,y) = I(x,y) \odot S(x,y) \odot G(x,y). \tag{5}$$

An example of a Gaussian shadow is shown in Fig. 6.

### 4.4 Speckle Reduction

Speckle noise is caused by interference between ultrasound waves. The speckle pattern observed in images captured using machines from different vendors often differs due to image enhancement and various filtering methods. As described in Ostvik et al. (2021), we apply a bilateral filter with randomly sampled parameter values to reduce the effect of these speckle patterns. We use the bilateral filter implementation from scikit-image (van der Walt et al., 2014). The $\sigma_{\text{spatial}}$ and $\sigma_{\text{color}}$ are sampled uniformly from the ranges $[0.1, 2.0)$ and $[0, 1)$, respectively. An example of this augmentation is shown in Fig. 7.

## 5 Evaluating the Efficacy of Individual Augmentations

We begin by addressing two fundamental questions: (a) how effective is each augmentation when applied individually? and (b) how does their effectiveness vary across different domains and tasks?

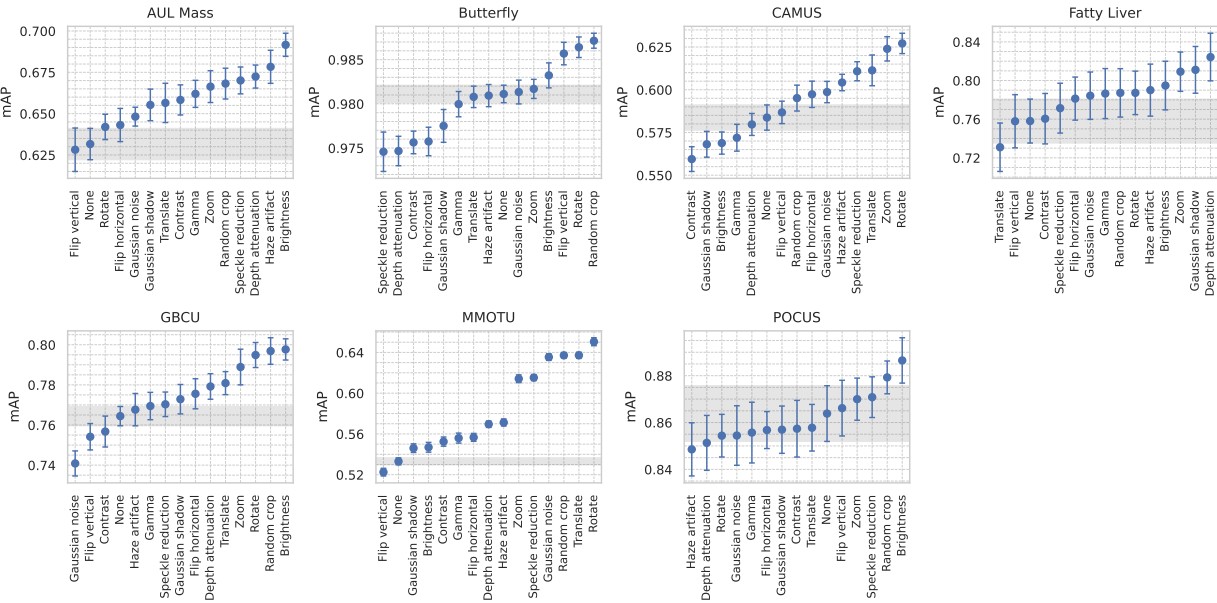

Figure 8: The mean and standard error of the mean average precision (mAP) using each augmentation as well as without augmentation (None) on each classification task. The shaded area highlights the standard error for the estimate of the mean mAP without data augmentation.

In these experiments, we compare the effectiveness of the top 10 most popular augmentations identified in Section 2.1: flipping (horizontal and vertical separately), rotation, resizing, random cropping, translation (along the $x$ and $y$ axes), contrast adjustment, brightness adjustment, Gaussian noise, Gamma adjustment; and the four ultrasound-specific data augmentations we previously described: depth attenuation, haze artifact addition, Gaussian shadow, and speckle reduction.

## 5.1 Evaluation Protocol

We evaluated the augmentations on each of the 14 classification and segmentation tasks described in Section 3. In each case, we fine-tuned models using each augmentation in isolation that had been pre-trained on ImageNet. For classification tasks, we used EfficientNetB0 (Tan & Le, 2019) models, while for segmentation tasks we used UNet models (Ronneberger et al., 2015) with EfficientNetB0 backbones. We used EfficientNets for both sets of tasks because of their popularity in medical imaging and to maintain consistency. We used the smallest B0 variants to make it feasible to run the evaluations across all tasks. Additional results on a subset of tasks using larger EfficientNet-B5 and Transformer models are presented in Appendix E. We used model implementations and pre-trained checkpoints from the MONAI library (The MONAI Consortium, 2020).

During training, we applied the augmentations with random strength (where applicable) and with 50 % probability on each image in an online fashion. The strength ranges for each augmentation were not tuned specifically for each task. Instead, a sensible range was chosen that is consistent with prior works. The parameters of each augmentation are listed in Appendix B. The images were normalized and resized so that the longest edge measured 224 px and padded (if needed) so that the final image measured $224 \times 224$ px before applying data augmentation. The only exception was when using random crop, in which case the images were resized to $256 \times 256$ px before being cropped to $224 \times 224$ px.

For each task, we performed 30 training runs using different random seeds for each augmentation. We measured performance on the classification tasks using the mean average precision (mAP) and on the segmentation tasks using the mean Dice score across all classes. Because the area(s) of interest in the images are small, we omitted the background class when calculating the Dice scores so that performance was not

| Augmentation | Task | | | | | | | | | | | | | | |
|---|---|---|---|---|---|---|---|---|---|---|---|---|---|---|---|
| | AUL Mass | | Butterfly | | CAMUS | | Fatty Liver | | GBCU | | MMOTU | | POCUS | | Mean |
| | mAP | Δ% | mAP | Δ% | mAP | Δ% | mAP | Δ% | mAP | Δ% | mAP | Δ% | mAP | Δ% | Δ% |
| None | 0.632 ± 0.009 | | 0.981 ± 0.001 | | 0.584 ± 0.007 | | 0.758 ± 0.023 | | 0.764 ± 0.005 | | 0.533 ± 0.003 | | 0.864 ± 0.012 | | |
| *Photometric* | | | | | | | | | | | | | | | |
| Speckle reduction | 0.670 ± 0.008 | +6.091 | 0.975 ± 0.002 | −0.666 | 0.611 ± 0.006 | +4.639 | 0.771 ± 0.026 | +1.761 | 0.770 ± 0.006 | +0.772 | 0.615 ± 0.003 | +15.412 | 0.871 ± 0.009 | +0.810 | +4.12 |
| Brightness | 0.692 ± 0.007 | +9.504 | 0.983 ± 0.001 | +0.215 | 0.569 ± 0.006 | −2.558 | 0.795 ± 0.025 | +4.846 | 0.798 ± 0.005 | +4.349 | 0.547 ± 0.005 | +2.590 | 0.887 ± 0.010 | +2.626 | +3.08 |
| Contrast | 0.658 ± 0.009 | +4.225 | 0.976 ± 0.001 | −0.559 | 0.559 ± 0.007 | −4.163 | 0.760 ± 0.026 | +0.321 | 0.757 ± 0.008 | −1.005 | 0.552 ± 0.004 | +3.635 | 0.857 ± 0.012 | −0.749 | +0.24 |
| Depth attenuation | 0.672 ± 0.007 | +6.462 | 0.975 ± 0.002 | −0.658 | 0.580 ± 0.006 | −0.693 | 0.824 ± 0.025 | +8.738 | 0.779 ± 0.006 | +1.934 | 0.570 ± 0.003 | +6.839 | 0.851 ± 0.012 | −1.448 | +3.02 |
| Gamma | 0.662 ± 0.008 | +4.808 | 0.980 ± 0.001 | −0.118 | 0.572 ± 0.008 | −2.026 | 0.786 ± 0.026 | +3.751 | 0.769 ± 0.007 | +0.659 | 0.556 ± 0.005 | +4.279 | 0.856 ± 0.013 | −0.941 | +1.49 |
| Gaussian noise | 0.648 ± 0.006 | +2.623 | 0.981 ± 0.001 | +0.022 | 0.599 ± 0.006 | +2.555 | 0.784 ± 0.025 | +3.465 | 0.741 ± 0.006 | −3.084 | 0.635 ± 0.003 | +19.191 | 0.854 ± 0.013 | −1.085 | +3.38 |
| Gaussian shadow | 0.655 ± 0.010 | +3.748 | 0.978 ± 0.002 | −0.368 | 0.568 ± 0.008 | −2.678 | 0.811 ± 0.024 | +6.990 | 0.773 ± 0.007 | +1.104 | 0.546 ± 0.004 | +2.427 | 0.857 ± 0.010 | −0.797 | +1.49 |
| Haze artifact | 0.678 ± 0.010 | +7.396 | 0.981 ± 0.001 | −0.019 | 0.604 ± 0.005 | +3.510 | 0.790 ± 0.027 | +4.222 | 0.768 ± 0.008 | +0.422 | 0.571 ± 0.003 | +7.153 | 0.849 ± 0.011 | −1.770 | +2.99 |
| *Geometric* | | | | | | | | | | | | | | | |
| Flip H. | 0.643 ± 0.010 | +1.823 | 0.976 ± 0.002 | −0.548 | 0.597 ± 0.008 | +2.330 | 0.781 ± 0.022 | +3.066 | 0.776 ± 0.007 | +1.457 | 0.557 ± 0.004 | +4.416 | 0.857 ± 0.008 | −0.816 | +1.68 |
| Flip V. | 0.628 ± 0.013 | −0.544 | 0.986 ± 0.001 | +0.466 | 0.587 ± 0.007 | +0.516 | 0.758 ± 0.028 | −0.028 | 0.754 ± 0.007 | −1.343 | 0.523 ± 0.004 | −1.985 | 0.866 ± 0.012 | +0.269 | −0.38 |
| Random crop | 0.668 ± 0.009 | +5.788 | 0.987 ± 0.001 | +0.614 | 0.595 ± 0.008 | +1.950 | 0.787 ± 0.025 | +3.847 | 0.797 ± 0.007 | +4.246 | 0.637 ± 0.003 | +19.506 | 0.879 ± 0.007 | +1.788 | +5.39 |
| Rotate | 0.642 ± 0.008 | +1.638 | 0.986 ± 0.001 | +0.539 | 0.627 ± 0.006 | +7.429 | 0.787 ± 0.022 | +3.848 | 0.795 ± 0.006 | +3.984 | 0.650 ± 0.004 | +22.018 | 0.854 ± 0.010 | −1.093 | +5.48 |
| Translate | 0.656 ± 0.012 | +3.940 | 0.981 ± 0.001 | −0.033 | 0.611 ± 0.009 | +4.727 | 0.731 ± 0.025 | −3.576 | 0.781 ± 0.006 | +2.153 | 0.637 ± 0.003 | +19.519 | 0.858 ± 0.010 | −0.700 | +3.72 |
| Zoom | 0.666 ± 0.010 | +5.501 | 0.982 ± 0.001 | +0.058 | 0.624 ± 0.007 | +6.886 | 0.809 ± 0.020 | +6.742 | 0.789 ± 0.009 | +3.202 | 0.614 ± 0.004 | +15.209 | 0.870 ± 0.009 | +0.712 | +5.47 |

Table 1: Mean average precision (mAP) for each augmentation across classification tasks. We report the mean ± standard error and relative improvement (Δ%) for individual tasks and averaged across all tasks.

exaggerated. Before the evaluations, we tuned the learning rate, weight decay, length of training in epochs, and dropout rate per task without data augmentation using Optuna (Akiba et al., 2019). Each model was trained for a minimum of 100 epochs and all model's converged within the allotted training budget, even with data augmentation. More details of this procedure are provided in Appendix C.

## 5.2 Classification Results

Fig. 8 compares the models' mean average precision using each augmentation for each task. These values are also reported in Table 1 alongside the percentage improvement, both per-task and across all tasks.

Whether an augmentation increased performance depended on the task. Zoom and random crop were the only augmentations that improved performance on all tasks, increasing performance on average by 5.47 % and 5.39 %, respectively. These were followed by rotate, brightness and speckle reduction that were each effective on 6/7 tasks. At the other end, contrast adjustment and vertical flip were the least effective. They produced the lowest average performance gains (+0.24 % and −0.38 %, respectively) and improved performance on the lowest number of tasks (3/7). Vertical flip was the only augmentation to slightly decrease performance on average (−0.38 %).

There was a clear divide between the efficacy of photometric and geometric transforms. Despite being detrimental to performance on the POCUS task (-1.09 %), rotate produced the highest average performance gains (+5.48 %), followed by zoom (+5.47 %) and random cropping (+5.39 %). In contrast, the most effective photometric augmentation, speckle reduction (+4.12 %) only ranked fourth overall.

While not the most effective, the ultrasound-specific augmentations (speckle reduction, depth attenuation, haze artifact and Gaussian shadow) were still reasonably effective. They were beneficial on 4–6 tasks and produced gains of 1.49 %–4.12 % on average. They were also very effective on some tasks. For example, depth attenuation ranked first on the Fatty Liver task, improving performance by 8.74 %.

## 5.3 Segmentation Results

Figure 9 compares the models' dice scores using each augmentation for each task. These values are also reported in Table 2 alongside the percentage improvement, both per-task and across all tasks.

In general, the relative improvements on segmentation tasks were much smaller than on the classification tasks. The average improvement of the top performing augmentation (brightness) was only 2.03 %, which would rank only tenth on the classification tasks. It also introduced substantial variability between runs. In fact, in many cases the gains were practically insignificant, with performance increases of < 1 %. We discuss this more deeply in Section 7.1. Unlike for the classification tasks, we did not observe a clear divide between the geometric and photometric augmentations. Brightness was the only augmentation to improve

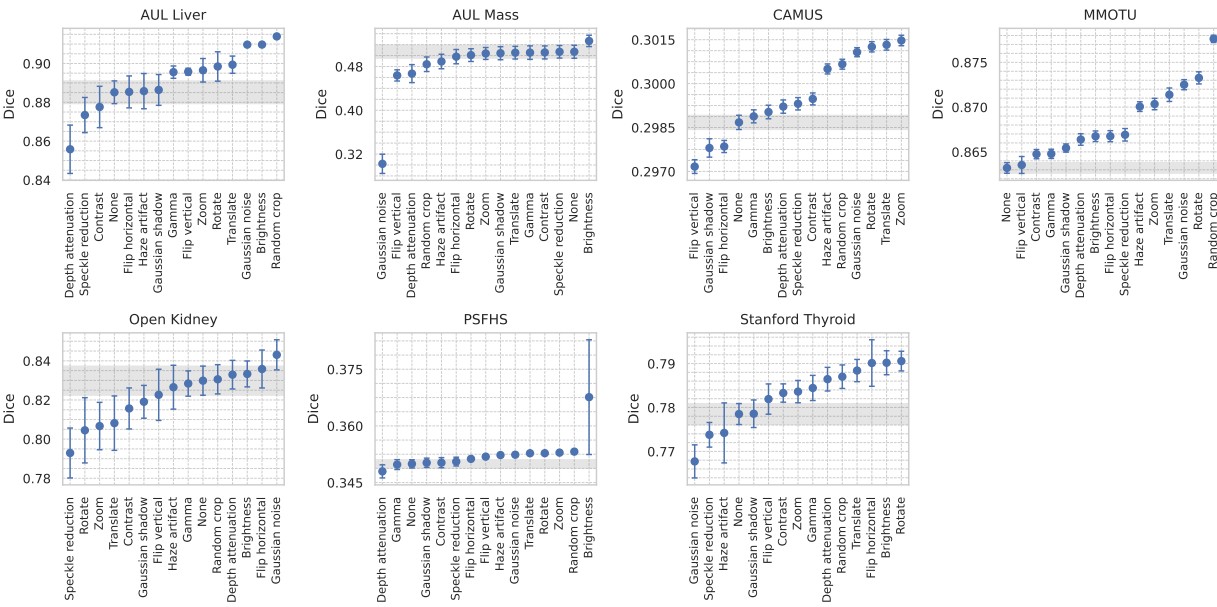

Figure 9: The mean and standard error of the mean dice score using each augmentation as well as without augmentation (None) on each segmentation task. The shaded area highlights the standard error for the estimate of the mean dice score without data augmentation.

performance on all tasks, while random crop improved performance on six and Gaussian noise, horizontal flip, rotate, translate, and zoom all improved performance on five tasks.

We also observed marked differences in the effectiveness of augmentations between different tasks on the same dataset. For example, Gaussian noise was moderately effective for the AUL liver segmentation task ($+2.77\,\%$), but was very detrimental to performance on the mass segmentation task ($-40.41\,\%$). Furthermore, while 11 augmentations were beneficial for liver segmentation only a single augmentation was beneficial for liver mass segmentation. These differences highlight the important influence of the task on the efficacy of different augmentations.

## 5.4 Comparing the Classification and Segmentation Results

The tasks on the AUL, CAMUS, and MMOTU datasets allow us to directly compare the efficacy of augmentations across classification and segmentation tasks using the same data. On the AUL tasks, we observed that fewer augmentations were effective as the difficulty of the task increased. 13 augmentations improved performance for mass classification compared to 11 for liver segmentation and only one for liver mass segmentation.

On the CAMUS tasks, a similar number of augmentations were beneficial for classification (11) and segmentation (9). However, some augmentations were only beneficial on one and not the other. Both vertical and horizontal flipping were beneficial for image quality classification, but not for cardiac region segmentation. Perhaps this is because these augmentations create anatomically incorrect images, which is important for region segmentation. On the other hand, brightness, contrast adjustment, depth attenuation and gamma adjustment were all beneficial for region segmentation, but not for image quality classification. Again, perhaps linked with the effect these attributes have on subjective image quality. Finally, on the MMOTU tasks all augmentations were effective on both tasks with the exception of vertical flipping. This was detrimental to classification performance and had little to no effect on segmentation performance.

These results demonstrate that even in controlled settings, there is much variation in performance between domains, tasks, and datasets. Which augmentations were or were not useful varied depending on the domain

| | Task | | | | | | | | | | | | | | |
|---|---|---|---|---|---|---|---|---|---|---|---|---|---|---|---|
| | AUL Liver | | AUL Mass | | CAMUS | | MMOTU | | Open Kidney | | PSFHS | | Stanford Thyroid | | Mean |
| Augmentation | Dice | Δ% | Dice | Δ% | Dice | Δ% | Dice | Δ% | Dice | Δ% | Dice | Δ% | Dice | Δ% | Δ% |
| None | $0.885 \pm 0.006$ | | $0.507 \pm 0.012$ | | $0.299 \pm 0.000$ | | $0.863 \pm 0.001$ | | $0.830 \pm 0.007$ | | $0.350 \pm 0.001$ | | $0.779 \pm 0.002$ | | |
| *Photometric* | | | | | | | | | | | | | | | |
| Speckle reduction | $0.873 \pm 0.009$ | $-1.324$ | $0.507 \pm 0.011$ | $-0.009$ | $0.299 \pm 0.000$ | $+0.213$ | $0.867 \pm 0.001$ | $+0.429$ | $0.793 \pm 0.013$ | $-4.452$ | $0.351 \pm 0.001$ | $+0.172$ | $0.774 \pm 0.003$ | $-0.607$ | $-0.80$ |
| Brightness | $0.910 \pm 0.001$ | $+2.776$ | $0.527 \pm 0.010$ | $+3.909$ | $0.299 \pm 0.000$ | $+0.119$ | $0.867 \pm 0.001$ | $+0.410$ | $0.833 \pm 0.007$ | $+0.415$ | $0.368 \pm 0.015$ | $+5.047$ | $0.790 \pm 0.003$ | $+1.500$ | $+2.03$ |
| Contrast | $0.878 \pm 0.011$ | $-0.854$ | $0.506 \pm 0.012$ | $-0.241$ | $0.299 \pm 0.000$ | $+0.268$ | $0.865 \pm 0.001$ | $+0.178$ | $0.816 \pm 0.010$ | $-1.711$ | $0.350 \pm 0.001$ | $+0.089$ | $0.783 \pm 0.002$ | $+0.615$ | $-0.24$ |
| Depth attenuation | $0.856 \pm 0.070$ | $-3.316$ | $0.467 \pm 0.016$ | $-7.883$ | $0.299 \pm 0.000$ | $+0.181$ | $0.866 \pm 0.001$ | $+0.368$ | $0.833 \pm 0.007$ | $+0.370$ | $0.348 \pm 0.002$ | $-0.570$ | $0.786 \pm 0.003$ | $+1.021$ | $-1.40$ |
| Gamma | $0.896 \pm 0.003$ | $+1.172$ | $0.505 \pm 0.012$ | $-0.308$ | $0.299 \pm 0.000$ | $+0.069$ | $0.865 \pm 0.001$ | $+0.182$ | $0.828 \pm 0.006$ | $-0.176$ | $0.350 \pm 0.001$ | $-0.055$ | $0.784 \pm 0.003$ | $+0.763$ | $-0.24$ |
| Gaussian noise | $0.910 \pm 0.001$ | $+2.772$ | $0.302 \pm 0.018$ | $-40.406$ | $0.301 \pm 0.000$ | $+0.803$ | $0.873 \pm 0.001$ | $+1.077$ | $0.843 \pm 0.008$ | $+1.591$ | $0.352 \pm 0.000$ | $+0.692$ | $0.768 \pm 0.004$ | $-1.385$ | $-4.98$ |
| Gaussian shadow | $0.886 \pm 0.008$ | $+0.137$ | $0.504 \pm 0.012$ | $-0.525$ | $0.298 \pm 0.000$ | $-0.292$ | $0.865 \pm 0.000$ | $+0.257$ | $0.819 \pm 0.008$ | $-1.301$ | $0.350 \pm 0.001$ | $+0.081$ | $0.779 \pm 0.003$ | $+0.009$ | $-0.23$ |
| Haze artifact | $0.886 \pm 0.009$ | $+0.067$ | $0.489 \pm 0.013$ | $-3.516$ | $0.301 \pm 0.000$ | $+0.611$ | $0.870 \pm 0.001$ | $+0.794$ | $0.827 \pm 0.011$ | $-0.402$ | $0.352 \pm 0.000$ | $+0.666$ | $0.774 \pm 0.007$ | $-0.552$ | $-0.33$ |
| *Geometric* | | | | | | | | | | | | | | | |
| Flip H. | $0.885 \pm 0.008$ | $+0.022$ | $0.498 \pm 0.013$ | $-1.761$ | $0.298 \pm 0.000$ | $-0.276$ | $0.867 \pm 0.001$ | $+0.411$ | $0.836 \pm 0.010$ | $+0.718$ | $0.351 \pm 0.000$ | $+0.377$ | $0.790 \pm 0.005$ | $+1.493$ | $-0.14$ |
| Flip V. | $0.896 \pm 0.002$ | $+1.195$ | $0.464 \pm 0.010$ | $-8.535$ | $0.297 \pm 0.000$ | $-0.505$ | $0.864 \pm 0.001$ | $+0.038$ | $0.823 \pm 0.013$ | $-0.872$ | $0.352 \pm 0.000$ | $+0.546$ | $0.782 \pm 0.003$ | $+0.438$ | $-1.10$ |
| Random crop | $0.914 \pm 0.001$ | $+3.252$ | $0.484 \pm 0.013$ | $-4.504$ | $0.301 \pm 0.000$ | $+0.666$ | $0.878 \pm 0.000$ | $+1.670$ | $0.831 \pm 0.007$ | $+0.085$ | $0.353 \pm 0.000$ | $+0.925$ | $0.787 \pm 0.003$ | $+1.093$ | $+0.46$ |
| Rotate | $0.898 \pm 0.008$ | $+1.501$ | $0.501 \pm 0.012$ | $-1.211$ | $0.301 \pm 0.000$ | $+0.864$ | $0.873 \pm 0.001$ | $+1.165$ | $0.805 \pm 0.017$ | $-3.055$ | $0.353 \pm 0.000$ | $+0.807$ | $0.791 \pm 0.002$ | $+1.549$ | $-0.23$ |
| Translate | $0.899 \pm 0.004$ | $+1.605$ | $0.505 \pm 0.012$ | $-0.332$ | $0.301 \pm 0.000$ | $+0.888$ | $0.871 \pm 0.001$ | $+0.947$ | $0.808 \pm 0.014$ | $-2.615$ | $0.353 \pm 0.000$ | $+0.803$ | $0.788 \pm 0.003$ | $+1.272$ | $+0.37$ |
| Zoom | $0.896 \pm 0.006$ | $+1.281$ | $0.504 \pm 0.011$ | $-0.614$ | $0.301 \pm 0.000$ | $+0.937$ | $0.870 \pm 0.001$ | $+0.827$ | $0.807 \pm 0.012$ | $-2.795$ | $0.353 \pm 0.000$ | $+0.849$ | $0.784 \pm 0.003$ | $+0.656$ | $-0.16$ |

Table 2: Mean dice score for each augmentation across segmentation tasks. We report the mean ± standard error and relative improvement (Δ%) for individual tasks and averaged across all tasks.

(e.g., fetal vs. cardiac ultrasound), task type (classification vs. segmentation) and the particular task being performed. Further experiments using larger EfficientNet and transformer models (Appendix E) show that the choice of model also impacts the efficacy of different augmentations. In the following section, we investigate the possibility of extracting greater, more consistent performance gains through more diverse augmentation.

# 6    Applying Multiple Augmentations

In the previous section, we demonstrated that individual data augmentations can improve model generalization on ultrasound image analysis tasks. However, in practice, we typically combine multiple augmentations to increase data diversity in different ways. This raises the question of how we should combine them. The AutoAugment family of algorithms (Cubuk et al., 2019; Lim et al., 2019; Hataya et al., 2019) pioneered automated augmentation strategy optimization, but these methods are computationally expensive and data-intensive. This has limited their adoption beyond using the strategies discovered for natural image benchmarks, such as CIFAR-10 and ImageNet, in the original articles. Simpler alternatives such as RandAugment (Cubuk et al., 2020) and TrivialAugment (Müller & Hutter, 2021) have since demonstrated comparable performance on these tasks without costly optimization. However, these techniques remain underutilized in medical imaging. Extending our previous results, we investigate whether TrivialAugment is effective for ultrasound image analysis, and examine how the inclusion of different augmentations affects performance.

## 6.1    Evaluation Protocol

TrivialAugment transforms each image by randomly selecting two augmentations with replacement (including the possibility of no augmentation) from a predefined set, and applies them sequentially. To evaluate how performance changes as we expand beyond the individually most effective augmentations, we trained separate models using TrivialAugment with the top-$N$ most effective augmentations for each task, where $N$ ranged from 2 to 14. These augmentation sets correspond to reading right-to-left across the sub-figures in figures 8 and 9. All other experimental conditions matched those used in our individual augmentation evaluations in Section 5.1.

## 6.2    Classification Results

Fig. 10 displays the trends in performance as the set of augmentations is expanded for each classification task. The percentage improvements, both per-task and across all tasks, are reported in Table 4 of Appendix D. Across all tasks, the best TrivialAugment configuration outperformed both the no augmentation and single best augmentation baselines. The best configurations produced a further 0.38 %–9.02 % improvement in mean average precision over the single best augmentation and 0.99 %–30.26 % improvement over no augmentation.

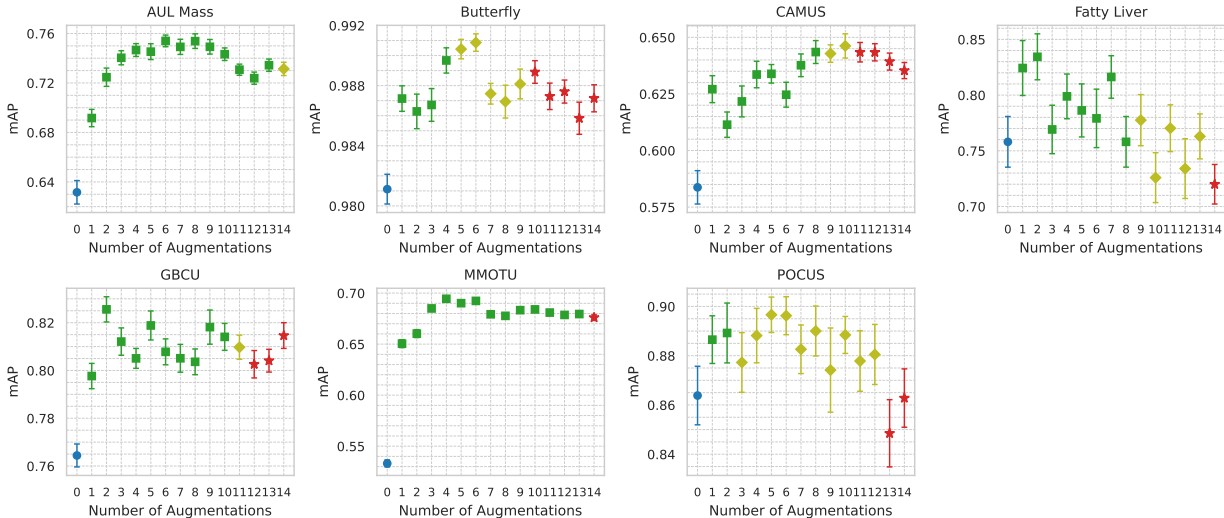

Figure 10: The mean and standard error of the mean average precision (mAP) using the Top-$N$ augmentations on the classification tasks. ● shows the performance without data augmentation. ■, ◆ and ★ show the addition of effective, ineffective and harmful augmentations, respectively.

For each task, performance initially increased as augmentations were added before declining. On the CAMUS, Butterfly, and POCUS tasks, the decline in performance coincided with the addition of individually harmful augmentations. However, on the remaining tasks performance began declining well before incorporating harmful augmentations. We discuss possible explanations for this in Section 7.3.

Finally, TrivialAugment still surpassed the no augmentation baseline across most tasks using all 14 augmentations, i.e., without requiring curated augmentation sets. The only exceptions were the Fatty Liver and POCUS tasks. This shows we can still achieve performance gains without investing resources in evaluating and optimizing individual augmentations and the augmentation set.

## 6.3   Segmentation Results

Figure 11 shows the trends in performance as the set of augmentations is expanded for each segmentation task. The percentage improvements, both per-task and across all tasks, are reported in Table 5 of Appendix D. As observed when evaluating the individual augmentations, the performance improvements are relatively modest in comparison to the gains seen in the classification tasks. While the best configurations produced 0.38 %–9.02 % improvements in the dice scores over no augmentation, the gains over the single best augmentation were between -0.01 % and 6.24 %, with the single best augmentation slightly outperforming TrivialAugment on the AUL Mass and PSFHS tasks.

Across all tasks, we observe the same pattern as the classification tasks. Performance initially improves as the set of augmentations grows before declining. This pattern was also observed when repeating our experiments using larger EfficientNet and transformer models (Appendix E). The results on the CAMUS task are an exception where performance continues to increase, with diminishing returns, as augmentations are added – even augmentations that reduced performance when evaluated individually. As observed on the classification tasks, we can link the declines on the AUL Liver, AUL Mass, and Open Kidney tasks with the introduction of harmful augmentations. With the exception of the AUL Mass segmentation task, TrivialAugment again consistently outperformed the no augmentation baseline even without careful augmentation selection.

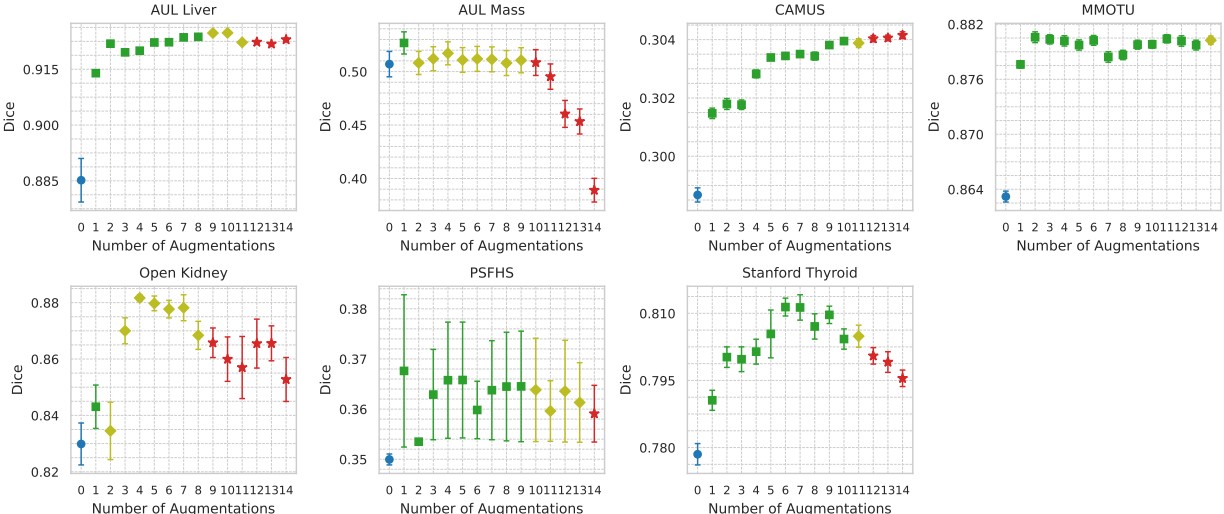

Figure 11: The mean and standard error of the dice score using the Top-$N$ augmentations on the segmentation tasks. ● shows the performance without data augmentation. ■, ◆ and ★ show the addition of effective, ineffective and harmful augmentations, respectively.

# 7 Discussion

Our experiments reveal both the potential and limitations of data augmentation for ultrasound analysis. We now examine the practical implications of our findings, present recommendations, and acknowledge important limitations of our study.

## 7.1 The Benefits of Augmentation for Classification and Segmentation Tasks

Despite low uptake, our experiments demonstrate that traditional domain-independent data augmentations are effective for ultrasound images and strongly support the use of data augmentation when training models for ultrasound image analysis tasks. This should dispel any notions that the generated images are unrealistic and therefore not useful. However, the improvements in the metrics are generally larger for classification tasks than for segmentation tasks, even when controlling for the dataset. This suggests several possibilities: there may be a lower ceiling for the effectiveness of data augmentation in segmentation, these tasks might be more sensitive to the strength of augmentation, or there is still room for developing more effective augmentations for segmentation tasks in medical imaging. Despite this, the small gains observed in some tasks should not be dismissed as insignificant. Ultimately, the practical significance of any gains depends on the specific use case, as performance is measured differently for different tasks and tolerances for error can vary.

## 7.2 Considerations for Domain and Modality-Specific Augmentations

Among the ultrasound-specific augmentations tested in our experiments, none consistently outperformed all traditional augmentations across tasks. Therefore, researchers should carefully evaluate whether these augmentations suffice before investing significant time and resources into developing custom augmentations for specific domains or tasks. When proposing new augmentations, their performance should be compared against existing augmentations to identify situations where they are beneficial.

Nevertheless, the four ultrasound-specific augmentations we tested still produced performance gains in most settings, and we demonstrated their broader effectiveness beyond the initial cardiac and nerve/blood vessel ultrasound contexts. Additionally, we hope our implementations lower the barrier to entry for using these augmentations and encourage further testing of their practical utility.

While we did not formally benchmark the computational cost of each augmentation, only speckle reduction increased the training time (by 1.5–2×) and only for the smaller models. This is despite the fact that the standard augmentations benefit from highly optimized implementations and our implementations weren't optimized for efficiency. When training the larger GPU-bottlenecked models (Appendix E) this difference was eradicated. Moreover, when using TrivialAugment the computational overhead of speckle reduction was negligible, even with small models, since each augmentation is selected relatively infrequently. These findings show that these ultrasound-specific techniques can be included without substantial computational cost in practice.

### 7.3 Considerations for TrivialAugment

The original analyses of TrivialAugment by Müller & Hutter (2021) on the CIFAR-10 dataset found that performance increased and then plateaued as the augmentation set size grew. However, our results across a wide range of ultrasound tasks demonstrate that careful selection of augmentations is crucial for maximizing performance in this setting. Not only did some augmentations harm performance, but for many tasks, performance declined after a certain point as more augmentations were added – even if they were effective individually. This decline might occur because each augmentation is applied less frequently as the set of augmentations grows, diluting the impact of more effective augmentations. A good approach is to evaluated each augmentation individually (i.e., training a single model per augmentation) to identify the most effective augmentations for the task. These can then be used with TrivialAugment to unlock better performance without having to invest the extensive effort required to design and tune a strategy (i.e., augmentation probabilities, strengths and sequences) either manually or via automated search. Even so, in 11 out of 14 tasks, blindly applying TrivialAugment without paring down the set of augmentations still led to improved performance and researchers should be encouraged by the fact that substantial gains can be achieved with limited tuning.

### 7.4 Limitations and Path Forward

In this work, we focused more on the individual effectiveness of different augmentations than different strategies for applying multiple augmentations. Despite that, we have demonstrated the importance of removing ineffective augmentations from the augmentation set when using TrivialAugment on ultrasound images to achieve peak performance. The extent to which these results hold for different model architectures and training algorithms is likely to vary, but we expect the general conclusions to remain the same. Moreover, we provide strong evidence for the use of data augmentation in training ultrasound analysis models and hope that researchers reconsider the omission of data augmentation in future work. Finally, we hope that our findings and methodology motivate and inform follow-up studies on other medical imaging modalities.

## 8 Conclusions

In this work we addressed a gap in our knowledge of effective data augmentation for ultrasound images, conducting the most rigorous analysis to date of commonly used data augmentations for ultrasound image analysis and comparing them against augmentations specifically designed for ultrasound scans. The results demonstrate that strong performance gains are possible using data augmentation, both when applied individually and when combined using TrivialAugment. As part of our contributions, we created a standardized benchmark for ultrasound image analysis tasks, reducing the effort required to evaluate ultrasound image analysis methods and allowing researchers to test their methods more broadly across a wider range of tasks and domains. We hope that these tools and findings encourage researchers and practitioners to use data augmentation more often and provide a blueprint for future investigations into the effectiveness of other data augmentation techniques or imaging modalities in medical imaging.

### Broader Impact Statement

This work aims to improve the reliability and accessibility of automated medical image analysis, particularly in settings where large datasets are difficult to obtain. While our techniques could enhance diagnostic

capabilities, they should complement rather than replace human oversight in clinical decision-making. We emphasize the importance of thorough validation across diverse patient populations to ensure that performance improvements translate equitably to different demographics and healthcare contexts.

**Acknowledgements**

This research was enabled in part by support provided by Calcul Québec (calculquebec.ca) and the Digital Research Alliance of Canada (alliancecan.ca). It was also supported through funding from Canadian Institute for Advanced Research (cifar.ca) and the Natural Sciences and Engineering Research Council of Canada (nserc-crsng.gc.ca).

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

# A  Examples of Generated Ultrasound Scan Masks

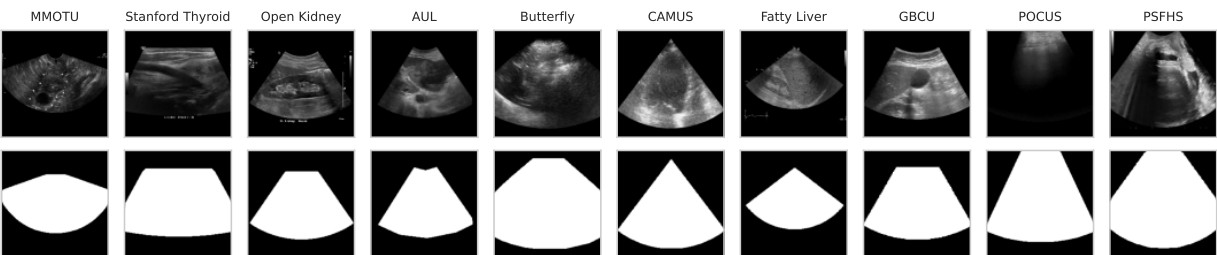

Figure 12: Examples of the ultrasound scan masks generated for each dataset in UltraBench.

## B   Augmentation Parameters

Table 3 contains the parameters for each of the augmentations tested in our experiment.

| Augmentation | Albumentations Class | Parameters |
|---|---|---|
| Bilateral filter | | $\sigma_{\text{spatial}} \in (0.05, 1.0)$ |
| | | $\sigma_{\text{spatial}} \in (0.05, 1.0)$ |
| | | window_size $= 5$ |
| Brightness | `RandomBrightnessContrast` | brightness_limit $\in (-0.2, 0.2)$ |
| Contrast | `RandomBrightnessContrast` | contrast_limit $\in (-0.2, 0.2)$ |
| Depth attenuation | | attenuation_rate $\in (0.0, 3.0)$ |
| | | max_attenuation $= 0.0$ |
| Flip vertical | `VerticalFlip` | N/A |
| Flip horizontal | `HorizontalFlip` | N/A |
| Gamma | `RandomGamma` | gamma_limit $\in (80, 120)$ |
| Gaussian noise | `GaussNoise` | var_limit $= 0.0225$ |
| | | mean $= 0.0$ |
| | | per_channel $=$ False |
| | | noise_scale_factor $= 1$ |
| Gaussian shadow | | strength $\in (0.25, 0.8)$ |
| | | $\sigma_x \in (0.01, 0.2)$ |
| | | $\sigma_y \in (0.01, 0.2)$ |
| Haze artifact | | radius $\in (0.05, 0.95)$ |
| | | $\sigma \in (0.0, 0.1)$ |
| Random crop | `RandomCrop` | width $= 224$ |
| | | height $= 224$ |
| Rotate | `Rotate` | limit $\in (-30, 30)$ |
| | | border_mode $= 0$ |
| | | value $= 0$ |
| Translate | `ShiftScaleRotate` | shift_limit $\in (-0.0625, 0.0625)$ |
| | | interpolation $= 1$ |
| | | border_mode $= 0$ |
| | | value $= 0$ |
| Zoom | `ShiftScaleRotate` | scale_limit $\in (-0.1, 0.1)$ |
| | | interpolation $= 1$ |
| | | border_mode $= 0$ |
| | | value $= 0$ |

Table 3: The settings for each of the augmentations tested in our experiments. We used the implementations from the Albumentations library where possible.

## C   Hyperparameter Tuning Procedure

To account for differences between tasks, in particular the number of examples in each dataset, we optimized the key regularization hyperparameters (training length in epochs, learning rate, dropout rates, and weight decay values) per task using the Optuna hyperparameter tuning framework. For each task, we performed 100 trials without using any data augmentation using the Tree-structured Parzen Estimator algorithm with the default parameter values. The values for the number of epochs were sampled from the set $\{50, 100, 200\}$, the learning rate sampled from the log domain between $(10^{-6}, 10^{-3})$, the dropout rates between $(0.0, 0.5)$, and the weight decay from the log domain between $(10^{-4}, 10^{-2})$. The optimized values for each hyperparameter are listed in the task configuration files in our accompanying source code.

# D TrivialAugment Results

| Strategy | AUL Mass | | Butterfly | | CAMUS | | Fatty Liver | | GBCU | | MMOTU | | POCUS | | Mean |
| | mAP | Δ% | mAP | Δ% | mAP | Δ% | mAP | Δ% | mAP | Δ% | mAP | Δ% | mAP | Δ% | Δ% |
|---|---|---|---|---|---|---|---|---|---|---|---|---|---|---|---|
| None | $0.632 \pm 0.009$ | | $0.981 \pm 0.001$ | | $0.584 \pm 0.007$ | | $0.758 \pm 0.023$ | | $0.764 \pm 0.005$ | | $0.533 \pm 0.003$ | | $0.864 \pm 0.012$ | | |
| *Top-N* | | | | | | | | | | | | | | | |
| 1 | $0.692 \pm 0.007$ | +9.504 | $0.987 \pm 0.001$ | +0.614 | $0.627 \pm 0.006$ | +7.429 | $0.824 \pm 0.025$ | +8.738 | $0.798 \pm 0.005$ | +4.349 | $0.650 \pm 0.004$ | +22.018 | $0.887 \pm 0.010$ | +2.626 | +7.90 |
| 2 | $0.725 \pm 0.007$ | +14.729 | $0.986 \pm 0.001$ | +0.527 | $0.611 \pm 0.006$ | +4.742 | $0.834 \pm 0.021$ | +10.066 | $0.826 \pm 0.005$ | +8.000 | $0.660 \pm 0.004$ | +23.856 | $0.889 \pm 0.012$ | +2.940 | +9.27 |
| 3 | $0.740 \pm 0.006$ | +17.218 | $0.987 \pm 0.001$ | +0.571 | $0.622 \pm 0.007$ | +6.501 | $0.769 \pm 0.022$ | +1.466 | $0.812 \pm 0.006$ | +6.237 | $0.685 \pm 0.003$ | +28.485 | $0.877 \pm 0.012$ | +1.556 | +8.86 |
| 4 | $0.747 \pm 0.005$ | +18.209 | $0.990 \pm 0.001$ | +0.873 | $0.634 \pm 0.006$ | +8.538 | $0.799 \pm 0.020$ | +5.381 | $0.805 \pm 0.004$ | +5.317 | $0.694 \pm 0.002$ | +30.259 | $0.888 \pm 0.011$ | +2.818 | +10.20 |
| 5 | $0.745 \pm 0.006$ | +18.002 | $0.990 \pm 0.001$ | +0.949 | $0.634 \pm 0.004$ | +8.591 | $0.786 \pm 0.024$ | +3.720 | $0.819 \pm 0.006$ | +7.119 | $0.690 \pm 0.002$ | +29.454 | $0.897 \pm 0.007$ | +3.796 | +10.23 |
| 6 | $0.754 \pm 0.005$ | +19.376 | $0.991 \pm 0.001$ | +0.994 | $0.625 \pm 0.005$ | +7.008 | $0.779 \pm 0.026$ | +2.776 | $0.808 \pm 0.005$ | +5.675 | $0.692 \pm 0.002$ | +29.874 | $0.896 \pm 0.008$ | +3.751 | +9.92 |
| 7 | $0.749 \pm 0.006$ | +18.616 | $0.987 \pm 0.001$ | +0.647 | $0.638 \pm 0.005$ | +9.237 | $0.816 \pm 0.019$ | +7.679 | $0.805 \pm 0.006$ | +5.321 | $0.679 \pm 0.002$ | +27.410 | $0.883 \pm 0.010$ | +2.173 | +10.15 |
| 8 | $0.754 \pm 0.005$ | +19.348 | $0.987 \pm 0.001$ | +0.593 | $0.643 \pm 0.005$ | +10.238 | $0.758 \pm 0.023$ | +0.001 | $0.804 \pm 0.005$ | +5.128 | $0.678 \pm 0.002$ | +27.125 | $0.890 \pm 0.010$ | +3.033 | +9.35 |
| 9 | $0.749 \pm 0.006$ | +18.624 | $0.988 \pm 0.001$ | −0.548 | $0.643 \pm 0.004$ | +10.123 | $0.778 \pm 0.023$ | +2.569 | $0.818 \pm 0.007$ | +7.029 | $0.683 \pm 0.003$ | +28.172 | $0.874 \pm 0.017$ | +1.196 | +9.77 |
| 10 | $0.743 \pm 0.005$ | +17.677 | $0.989 \pm 0.001$ | +0.794 | $0.646 \pm 0.005$ | +10.700 | $0.726 \pm 0.022$ | −4.239 | $0.814 \pm 0.006$ | +6.494 | $0.684 \pm 0.002$ | +28.306 | $0.888 \pm 0.008$ | +2.847 | +8.94 |
| 11 | $0.731 \pm 0.004$ | +15.694 | $0.987 \pm 0.001$ | +0.629 | $0.643 \pm 0.004$ | +10.225 | $0.770 \pm 0.021$ | +1.613 | $0.810 \pm 0.005$ | +5.928 | $0.681 \pm 0.002$ | +27.727 | $0.878 \pm 0.012$ | +1.622 | +9.06 |
| 12 | $0.724 \pm 0.005$ | +14.609 | $0.988 \pm 0.001$ | +0.661 | $0.643 \pm 0.004$ | +10.219 | $0.734 \pm 0.027$ | −3.169 | $0.803 \pm 0.006$ | +4.997 | $0.679 \pm 0.003$ | +27.294 | $0.881 \pm 0.012$ | +1.931 | +8.08 |
| 13 | $0.734 \pm 0.005$ | +16.267 | $0.986 \pm 0.001$ | +0.639 | $0.611 \pm 0.004$ | +9.517 | $0.763 \pm 0.020$ | +0.643 | $0.804 \pm 0.005$ | +5.190 | $0.679 \pm 0.003$ | +27.456 | $0.848 \pm 0.014$ | −1.775 | +8.25 |
| 14 | $0.731 \pm 0.005$ | +15.794 | $0.987 \pm 0.001$ | +0.635 | $0.624 \pm 0.004$ | +8.837 | $0.720 \pm 0.018$ | −5.022 | $0.815 \pm 0.005$ | +6.566 | $0.676 \pm 0.003$ | +26.803 | $0.863 \pm 0.012$ | −0.119 | +7.64 |

Table 4: Mean average precision (mAP) for TrivialAugment with each set of the Top-$N$ augmentations across classification tasks. We report the mean ± standard error and relative improvement (Δ%) for individual tasks and averaged across all tasks.

| Strategy | AUL Liver | | AUL Mass | | CAMUS | | MMOTU | | Open Kidney | | PSFHS | | Stanford Thyroid | | Mean |
| | Dice | Δ% | Dice | Δ% | Dice | Δ% | Dice | Δ% | Dice | Δ% | Dice | Δ% | Dice | Δ% | Δ% |
|---|---|---|---|---|---|---|---|---|---|---|---|---|---|---|---|
| None | $0.885 \pm 0.006$ | | $0.507 \pm 0.012$ | | $0.299 \pm 0.000$ | | $0.863 \pm 0.001$ | | $0.830 \pm 0.007$ | | $0.350 \pm 0.001$ | | $0.779 \pm 0.002$ | | |
| *Top-N* | | | | | | | | | | | | | | | |
| 1 | $0.914 \pm 0.001$ | +3.252 | $0.527 \pm 0.010$ | +3.909 | $0.301 \pm 0.000$ | +0.937 | $0.878 \pm 0.000$ | +1.670 | $0.843 \pm 0.008$ | +1.591 | $0.368 \pm 0.015$ | +5.047 | $0.791 \pm 0.002$ | +1.549 | +2.57 |
| 2 | $0.922 \pm 0.000$ | +4.148 | $0.508 \pm 0.011$ | +0.222 | $0.302 \pm 0.000$ | +1.044 | $0.881 \pm 0.001$ | +2.014 | $0.835 \pm 0.010$ | +0.560 | $0.353 \pm 0.000$ | +1.013 | $0.800 \pm 0.002$ | +2.788 | +1.68 |
| 3 | $0.920 \pm 0.000$ | +3.885 | $0.512 \pm 0.011$ | +0.996 | $0.302 \pm 0.000$ | +1.036 | $0.880 \pm 0.001$ | +1.987 | $0.870 \pm 0.005$ | +4.838 | $0.363 \pm 0.009$ | +3.698 | $0.800 \pm 0.003$ | +2.728 | +2.74 |
| 4 | $0.920 \pm 0.000$ | +3.935 | $0.517 \pm 0.011$ | +1.994 | $0.303 \pm 0.000$ | +1.389 | $0.880 \pm 0.001$ | +1.967 | $0.882 \pm 0.001$ | +6.241 | $0.366 \pm 0.012$ | +4.518 | $0.801 \pm 0.003$ | +2.946 | +3.28 |
| 5 | $0.922 \pm 0.000$ | +4.184 | $0.511 \pm 0.012$ | +0.765 | $0.303 \pm 0.000$ | +1.578 | $0.880 \pm 0.001$ | +1.915 | $0.880 \pm 0.003$ | +6.010 | $0.366 \pm 0.012$ | +4.528 | $0.805 \pm 0.005$ | +3.454 | +3.20 |
| 6 | $0.922 \pm 0.000$ | +4.192 | $0.512 \pm 0.012$ | +0.967 | $0.303 \pm 0.000$ | +1.596 | $0.880 \pm 0.001$ | +1.973 | $0.878 \pm 0.003$ | +5.763 | $0.360 \pm 0.006$ | +2.821 | $0.811 \pm 0.002$ | +4.226 | +3.08 |
| 7 | $0.924 \pm 0.000$ | +4.339 | $0.511 \pm 0.012$ | +0.886 | $0.304 \pm 0.000$ | +1.617 | $0.878 \pm 0.001$ | +1.765 | $0.878 \pm 0.005$ | +5.823 | $0.364 \pm 0.010$ | +3.948 | $0.811 \pm 0.003$ | +4.214 | +3.23 |
| 8 | $0.924 \pm 0.000$ | +4.355 | $0.508 \pm 0.012$ | +0.199 | $0.303 \pm 0.000$ | +1.594 | $0.879 \pm 0.001$ | +1.789 | $0.868 \pm 0.005$ | +4.644 | $0.364 \pm 0.011$ | +4.155 | $0.807 \pm 0.003$ | +3.669 | +2.91 |
| 9 | $0.925 \pm 0.000$ | +4.470 | $0.511 \pm 0.012$ | +0.710 | $0.304 \pm 0.000$ | +1.721 | $0.880 \pm 0.000$ | +1.920 | $0.866 \pm 0.005$ | +4.326 | $0.365 \pm 0.011$ | +4.164 | $0.810 \pm 0.002$ | +4.002 | +3.04 |
| 10 | $0.925 \pm 0.000$ | +4.476 | $0.508 \pm 0.012$ | +0.290 | $0.304 \pm 0.000$ | +1.766 | $0.880 \pm 0.000$ | +1.924 | $0.860 \pm 0.008$ | +3.626 | $0.364 \pm 0.010$ | +3.965 | $0.804 \pm 0.002$ | +3.308 | +2.76 |
| 11 | $0.922 \pm 0.000$ | +4.187 | $0.495 \pm 0.012$ | −2.303 | $0.304 \pm 0.000$ | +1.743 | $0.880 \pm 0.000$ | +1.993 | $0.857 \pm 0.011$ | +3.268 | $0.360 \pm 0.006$ | +2.761 | $0.805 \pm 0.002$ | +3.390 | +2.15 |
| 12 | $0.922 \pm 0.000$ | +4.196 | $0.460 \pm 0.012$ | −9.183 | $0.304 \pm 0.000$ | +1.794 | $0.880 \pm 0.001$ | +1.966 | $0.865 \pm 0.009$ | +4.290 | $0.364 \pm 0.010$ | +3.892 | $0.800 \pm 0.002$ | +2.825 | +1.40 |
| 13 | $0.922 \pm 0.000$ | +4.139 | $0.453 \pm 0.012$ | −10.585 | $0.304 \pm 0.000$ | +1.802 | $0.880 \pm 0.001$ | +1.914 | $0.866 \pm 0.006$ | +4.302 | $0.361 \pm 0.008$ | +3.247 | $0.799 \pm 0.002$ | +2.646 | +1.07 |
| 14 | $0.923 \pm 0.000$ | +4.271 | $0.389 \pm 0.011$ | −23.256 | $0.304 \pm 0.000$ | +1.834 | $0.880 \pm 0.001$ | +1.975 | $0.853 \pm 0.008$ | +2.757 | $0.359 \pm 0.006$ | +2.608 | $0.795 \pm 0.002$ | +2.181 | −1.09 |

Table 5: Mean dice score for TrivialAugment with each set of the Top-$N$ augmentations across segmentation tasks. We report the mean ± standard error and relative improvement (Δ%) for individual tasks and averaged across all tasks.

# E Evaluations with Different Model Sizes and Architectures

To assess the influence of model size and architecture on augmentation effectiveness, we repeated our experiments using larger EfficientNet backbones, and Mix Transformer (MiT) and SegFormer models (Xie et al., 2021). We tested the individual augmentations and TrivialAugment on a subset of classification and segmentation tasks on three datasets: a small dataset (AUL Mass), a medium-size dataset (MMOTU) and a large dataset (CAMUS). For each task, we trained the models for 200 epochs and optimized the learning rate and weight decay per model on the validation set using a grid search over seven logarithmically spaced learning rates between $10^{-5}$ and $10^{-1}$ and seven logarithmically spaced weight decays between $10^{-6}$ and $10^{-3}$. All models reached convergence within 200 epochs.

## E.1 Individual Augmentations

Model size and architecture strongly influence augmentation rankings, just like the task and domain/dataset (Section 5). As shown in figures 13 and 14, the performance gains for each augmentation vary considerably between models, even when the pre-training data (ImageNet) is the same. This variation suggests that different models possess different levels of robustness to various characteristics, such as changes in brightness, which makes augmentations that perturb these characteristics less effective. Consequently, these findings

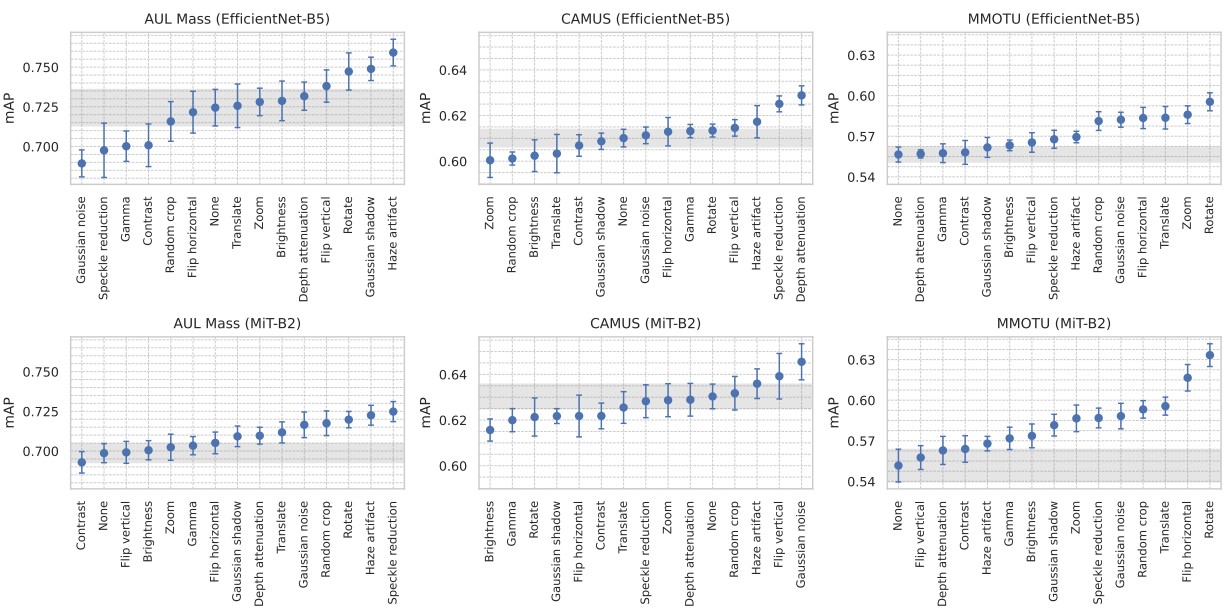

Figure 13: The mean and standard error of the mean average precision (mAP) using each augmentation as well as without augmentation (None) for different models on the AUL Mass, MMOTU and CAMUS classification tasks. The shaded area highlights the standard error for the estimate of the mean mAP without data augmentation.

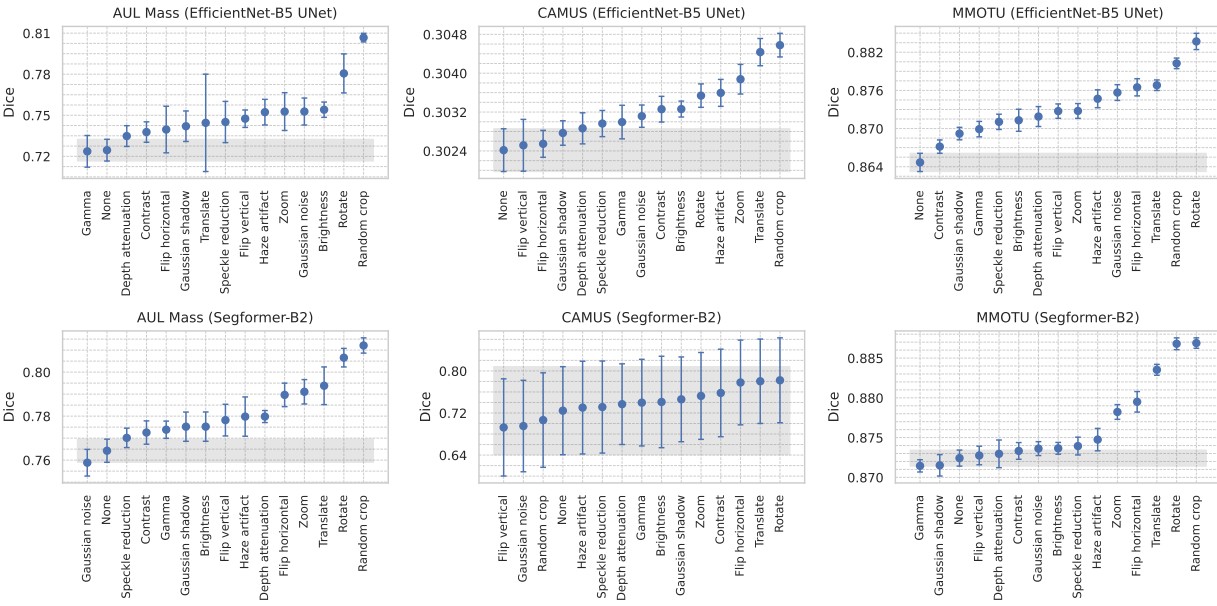

Figure 14: The mean and standard error of the mean dice score using each augmentation as well as without augmentation (None) for different models on the AUL Mass, MMOTU and CAMUS segmentation tasks. The shaded area highlights the standard error for the estimate of the mean dice score without data augmentation.

underscore the need to tailor data augmentation strategies to specific model architectures – much like tuning other hyperparameters, such as the learning rate.

### E.2    TrivialAugment

Our analysis reveals consistent trends in the performance of TrivialAugment across different models, despite variations in individual augmentation efficacy. As illustrated in figures 15 and 16, we classified augmentations into three categories: effective, ineffective, and harmful. This classification was based on whether the mean performance of each augmentation lies above, within, or below the standard error of the baseline (no augmentation). The results show the same performance pattern as our experiments across all the tasks with smaller models: utilizing only the most effective augmentations yields the highest model performance. Progressively introducing less effective or potentially harmful augmentations leads to a gradual decline in overall performance.

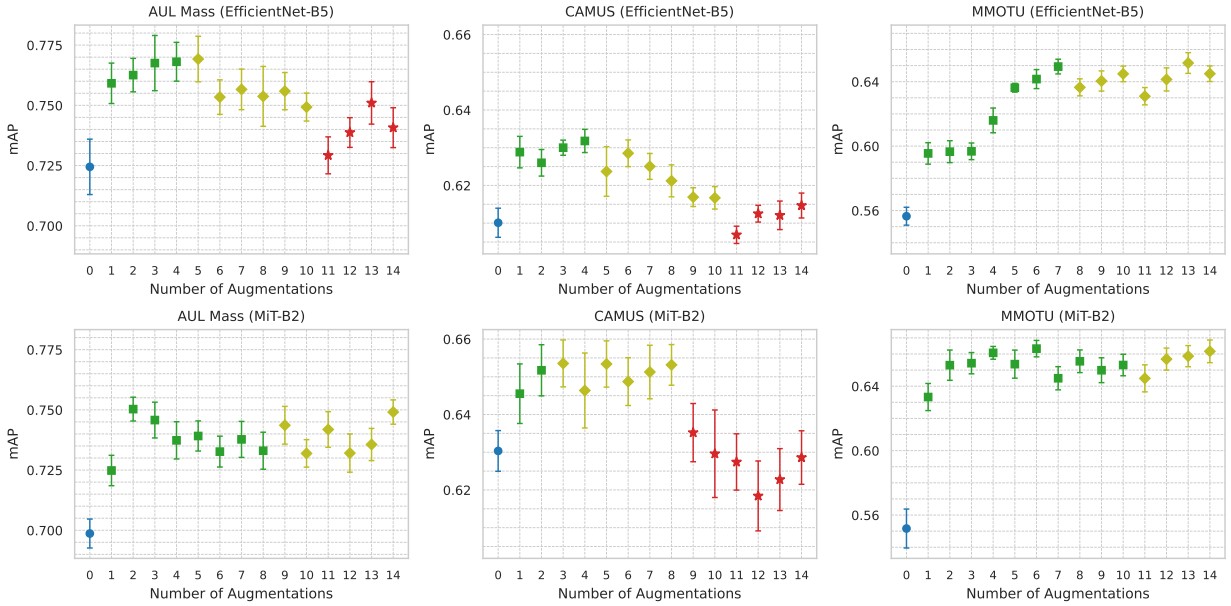

Figure 15: The mean and standard error of the mean average precision (mAP) using the Top-$N$ augmentations for different models on the AUL Mass, MMOTU and CAMUS classification tasks. ⬤ shows the performance without data augmentation. ■, ◆ and ★ show the addition of effective, ineffective and harmful augmentations, respectively.

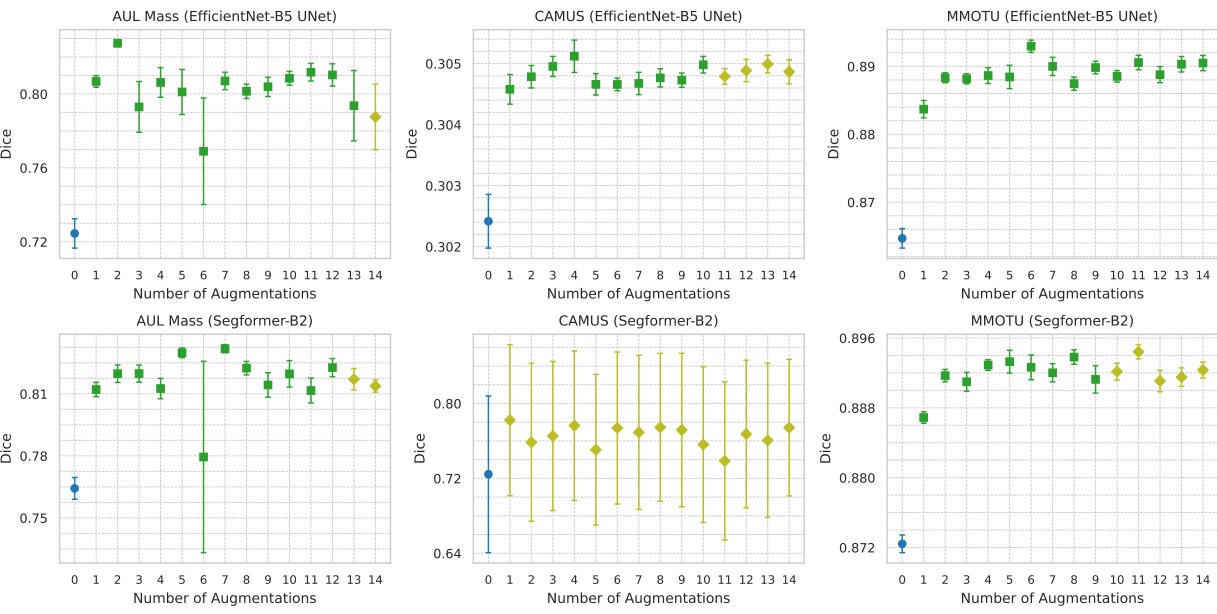

Figure 16: The mean and standard error of the dice score using the Top-$N$ augmentations for different models in the AUL Mass, MMOTU and CAMUS segmentation tasks. ● shows the performance without data augmentation. ■, ◆ and ⬟ show the addition of effective, ineffective and harmful augmentations, respectively.

