# OpenReview forum: "Revisiting Data Augmentation for Ultrasound Images"
_TMLR — Accepted by TMLR_

### Review · Reviewer_HpXm · 2025-03-03

**Summary Of Contributions:**

This paper evaluates the importance of data augmentation (DA) for ultrasound images. For doing that, the authors first introduce a new benchmark, which is the combination of several datasets of ultrasound images for image classification and semantic segmentation. Evaluations are performed for the most common general data augmentation as well as some more ultrasound specific augmentations. In addition also the combination of multiple augmentations is used through TrivialAumgent. Results show that general image augmentations are effective also on this domain and should be adopted by the community. In addition also modality specific augmentations could help. In addition, augmentations seems to be much more effective for image classification than segmentation. Finally, the combination of multiple augmentations with TrivialAugment can further improve performance.

**Audience:**

Yes

**Broader Impact Concerns:**

I am ok with that.

**Claims And Evidence:**

Yes

**Requested Changes:**

(critical) I would like to see the effect of DA on different models such as other CNN with different amount of parameters and architecture and ViTs. I would suggest to reduce the number of repetitions (to 3 or 5 for instance), but testing different models.

(critical) Although more expensive, the hyper-parameter tuning should be done for each specific augmentation to have optimal results. I would like to see results with these setting and compare with what proposed of tuning hyper-parms only on the training without DA.

(optional) It would be interesting to explore how the combination of multiple augmentation works. If you greedly select the best K augmentations is this optimal or there might be augmentations that are not compatible?

(optimal) It would be interesting to see what is the effect of pre-training in the proposed results.

**Strengths And Weaknesses:**

\+ The paper presents an interesting analysis of a topic that is well known for natural images, but not so much for specific domains, like in this case ultrasound images.

\+ To further motivate the importance of this work, authors show that for ultrasound images in most papers data augmentation is not really used, even if, with their results they show that it could effectively improve performance

\+ The paper proposes some simple strategies to boost performance on ultrasound images for image classification and segmentation without a large exploration of hyper-parameters

\- Results are validated only for EfficientNetB0. This is quite limited for a study of data augmentation. What is the effect of Data augmentation when changing the size of the model? The CNN architecture (e.g. ResNet, or ConvNext)? What about ViT? Would they perform similarly? I think those are common architectures and state-of-the art methods that cannot be overlooked.

\- It seems that the hyper-parameters of each model are optimized without data-augmentation (end of section 5.2). In my opinion, the hyper-parameters should be tuned with the augmentations. Consider for instance the number of training epochs. It is well-known that when adding DA the training would require more epochs to reach optimal performance.

\- The part of combinations of different augmentations could be explored more. Here authors consider using the simple TrivialAugment with the top-k best single augmentations. Without exploring other methods for combining the augmentations, did you consider that the combination of some augmentations can have some side effects. For instance, two augmentations could be incompatible and using them together could reduce the performance.

\- Some important details are missing. For instance, what kind of pre-training is used in the experiments? What is their effect? The trend would remain the same with a different or no pre-training?

\- At the bottom of page 4, authors talk about "convex" ultrasound images, without really clarifying the meaning of convex in this context.

---

> ### Author Response · Authors · 2025-05-30
> **Responses to your feedback and questions**
>
> Reviewer HpXm, thank you for your valuable feedback and suggestions. We hope that our additional results and clarifications address your questions.
>
> **We now have results on different model sizes and architectures**
>
> Following your suggestion, we have included results using larger EfficientNet-B5 (UNet) and transformer (MiT-B2/Segformer-B2 [10]) models for the AUL mass, MMOTU and CAMUS classification and segmentation tasks. These results (now included in Appendix E) show that Model size and architecture strongly influence augmentation ranking, just like the task and domain/dataset. However, the results using TrivialAugment show the same performance pattern as our experiments across all the tasks with smaller models: utilizing only the most effective augmentations yields the highest model performance and progressively introducing less effective or harmful augmentations leads to a gradual decline in overall performance.
>
> We selected only a single small (AUL mass), medium (MMOTU) and large (CAMUS) sized dataset as it is not possible for us to evaluate these models across the entire benchmark within our compute constraints.
>
> **Reasons for not performing hyperparameter tuning for each specific augmentation, task and model combination**
>
> You raise a valid point about the potential impact on the length of training with and without data augmentation. In fact, all models converged well before the maximum number of epochs in all conditions. We have added this clarification in our paper.
>
> Our choice to only tune the other hyperparameters (LR, weight decay, and dropout rate) per task and per model and reuse them for each condition (augmentation) is consistent with many other high quality studies of data augmentation [1-9]. It would be infeasible to tune these for every task, augmentation and model combination. For the individual evaluations alone (Sec. 5), this would require an additional $14 \times 14 = 196$ hyperparameter studies, each with 100 trials. The ablations with 4 different models on a subset of 6 tasks would require an additional $4 \times 6 \times 14 = 336$ studies.
>
> **The potential for incompatible augmentations when applying multiple augmentation**
>
> We appreciate the thoughtful question about augmentation compatibility. While exploring all possible augmentation combinations would indeed provide deeper insights, our primary goal was to demonstrate that substantial performance gains can be achieved without costly optimization processes.
>
> The combinatorial challenge of exhaustively testing augmentation interactions is prohibitively expensive. For our study, evaluating all pairwise combinations alone would require 14 tasks $\times {14\choose2}$ pairs $\times N$ repetitions $= 1274 \times N$ training runs, which is computationally infeasible.
>
> Even when certain augmentation pairs might be detrimental when applied together, TrivialAugment's probabilistic nature dilutes this negative impact. For example, in a pool of 5 augmentations where augmentations A and B are detrimental when combined (though each may be beneficial with other augmentations), this specific combination would only occur in approximately 5.56% of training examples (2 × ⅙ × ⅙, including the possibility of no augmentation). The positive impact of beneficial combinations likely outweighs these occasional harmful pairings.
>
> We conducted a preliminary investigation on the AUL mass classification task (our smallest dataset) early in our research, testing randomly sampled augmentation sets of different sizes. Using a leave-one-out approach, we observed that including or excluding any particular augmentation had negligible effect in most cases, particularly as the augmentation set size increased. This finding suggests that simple augmentation combination strategies like TrivialAugment with top-k selection provide a practical balance between performance improvement and computational efficiency.
>
> *Please see the following comment for responses to your remaining questions.*

---

> > ### Author Response · Authors · 2025-05-30
> > **Responses to your feedback and questions (cont.)**
> >
> > **The choice of using pre-trained models**
> >
> > As stated in Sec. 5.1, we consistently used ImageNet pre-training for all tasks, which is standard practice in medical image analysis due to limited data availability. While the reviewer raises an interesting question about how pre-training affects our results, there are several reasons we maintained this approach:
> >
> > - Pre-training was necessary to achieve acceptable performance on our smallest datasets
> > - Using pre-trained models standardized our experimental setup across all tasks
> > - This approach aligns with standard practices in the field
> >
> > For some medical imaging modalities like chest X-rays, large public datasets exist that could enable modality-specific pre-training, but ultrasound lacks such resources. While comparing different pre-training strategies (or no pre-training) could provide additional insights, such an experiment would substantially increase the computational cost of our study, requiring us to retrain all models across multiple datasets. Given the already extensive scope of our work (14 datasets, 14 augmentations, and multiple training strategies), we believe our current approach provides a fair and practical evaluation of augmentation techniques as they would be applied in real-world scenarios with limited data.
> >
> > **Additional references**
> >
> > [1] Cubuk et al., “AutoAugment: Learning Augmentation Strategies From Data,” in Proceedings of the IEEE/CVF Conference on Computer Vision and Pattern Recognition, Jun. 2019, pp. 113–123.
> >
> > [2] Müller and Hutter, “TrivialAugment: Tuning-Free Yet State-of-the-Art Data Augmentation,” in Proceedings of the IEEE/CVF International Conference on Computer Vision, 2021, pp. 774–782.
> >
> > [3] Ramakers et al., “UltraAugment: Fan-shape and Artifact-based Data Augmentation for 2D Ultrasound Images,” in Proceedings of the IEEE/CVF Conference on Computer Vision and Pattern Recognition, 2024, pp. 2422–2431.
> >
> > [4] Tirindelli et al., “Rethinking Ultrasound Augmentation: A Physics-Inspired Approach,” in Medical Image Computing and Computer Assisted Intervention – MICCAI 2021, 2021, pp. 690–700.
> >
> > [5] Singla et al., “Speckle and Shadows: Ultrasound-specific Physics-based Data Augmentation for Kidney Segmentation,” in Proceedings of The 5th International Conference on Medical Imaging with Deep Learning (MIDL 2022), Proceedings of Machine Learning Research: PMLR, 2022, pp. 1139--1148.
> >
> > [6] Yun et al., “CutMix: Regularization Strategy to Train Strong Classifiers With Localizable Features,” in Proceedings of the IEEE/CVF International Conference on Computer Vision, 2019, pp. 6023–6032.
> >
> > [7] Cubuk et al., “RandAugment: Practical Automated Data Augmentation with a Reduced Search Space,” in Advances in Neural Information Processing Systems, Curran Associates, Inc., 2020, pp. 18613–18624.
> >
> > [8] Raghu et al., “Data Augmentation for Electrocardiograms,” in Proceedings of the Conference on Health, Inference, and Learning, PMLR, Apr. 2022, pp. 282–310.
> >
> > [9] Zhang et al., “mixup: Beyond Empirical Risk Minimization,” in 6th International Conference on Learning Representations, ICLR 2018, April 30 - May 3, 2018, Conference Track Proceedings, 2018.
> >
> > [10] Xie et al., “SegFormer: Simple and Efficient Design for Semantic Segmentation with Transformers,” in Advances in Neural Information Processing Systems, Curran Associates, Inc., 2021, pp. 12077–12090.

---

### Review · Reviewer_LQtm · 2025-03-06

**Summary Of Contributions:**

The paper provides an exhaustive account of the different types of augmentations that have been applied for ultrasound images over the years. Furthermore, this paper also accounts for the different types of ultrasound images available and the augmentations optimal for them. It also studies the effectiveness of the different augmentations for different kinds of tasks as well.

**Audience:**

Yes

**Broader Impact Concerns:**

Addressed in the paper.

**Claims And Evidence:**

Yes

**Requested Changes:**

1. In connection with Weakness number 2, the authors can provide a comparison of the time required for the data processing pipeline with and without the augmentations in Sec 4.

2. If possible within the given period for revision, the authors could analyse the question in Weakness number 3.

**Strengths And Weaknesses:**

**Strengths**:

1. Conducts a rigorous analysis of the different types of augmentations used on ultrasound images.
2. Provides an in-depth comparison of the effect of different types of augmentations on performance.
3. The paper also covers a wide range of anatomical domains.

**Weakness**:

1. In Figure 8, it is clearly visible that the four augmentations that the authors claim to have not been used beyond their original works are not actually that good in improving performance over the wide range of tasks that the authors have selected.

For example, Randon Crop and Zoom figures in the top half for all the datasets, while none of the four, like Depth attenuation, Speckle reduction, Gaussian shadow, Haze artifact, does not.

How do the authors justify the choice of the augmentation in Section 4?

2. In continuation with the above point, does the performance improvement obtained using the augmentations in Section 4 provide a significant performance-complexity trade-off? The complexity in the previous sentence refers to the computational complexity of applying any augmentation mentioned in Section 4.

3. In most cases in Figure 8, we can see that the augmentations in Section 4 provide less performance improvement over several trivial augmentations like Random crop and zoom. Thus, I wonder if it would provide more insight if the TrivialAugment could be applied on the remaining 10 augmentations (without the augmentations mentioned in Section 4) and then the performance is analysed.

In other words, the authors can consider this as a question, whether combining the augmentations mentioned in Section 4 with the rest 10 truly improves performance.

---

> ### Author Response · Authors · 2025-05-30
> **Responses to your feedback and questions**
>
> Reviewer LQtm, thank you for careful analysis and insightful suggestions. We hope the following clarifications address your questions.
>
> **Choice of ultrasound-specific augmentations**
>
> Our choice of ultrasound (US)-specific augmentations (depth attenuation, speckle reduction, Gaussian shadow, and haze artifact) was guided by our literature review. These were the only US-specific augmentations that appeared in multiple studies, as noted in the opening paragraph of Sec. 4. In addition, these augmentations are also broadly applicable across different anatomical domains because they do not depend on anatomical features (e.g., the presence of bones), unlike some other US-specific augmentations discussed in Sec. 2.2. Their repeated use in the literature suggested potential effectiveness, yet their limited adoption across the broader ultrasound imaging community presented an interesting research question.
>
> **Performance of ultrasound-specific augmentations**
>
> The average performance improvements from the US-specific augmentations across all tasks are less than some standard techniques (e.g., random crop and zoom). However, importantly, there are certain tasks where this is not the case. One example of this is for the AUL mass classification task (Fig. 8, Table 1). Speckle reduction (+6.09%), depth attenuation (+6.46%), and haze artifact (+7.40%) all outperformed random crop (+5.79%) and zoom (+5.50%). This task-specific variation underscores our finding that augmentation efficacy is heavily dependent on the specific domain and task, which is an important takeaway.
>
> **Computational cost of ultrasound-specific augmentations**
>
> We did not formally benchmark the computational costs of different augmentations. However, during our evaluations of individual augmentations (Sec. 5) depth attenuation, haze artifact and Gaussian shadow had negligible impact on training time as compared to the standard augmentations and no data augmentation, while speckle reduction increased training time by 1.5–2x. This comparison has an important caveat: standard augmentations benefit from highly optimized implementations and our implementations weren’t optimized for efficiency. Furthermore, when training larger GPU-bottlenecked models (requested by Reviewer HpXm), none of the augmentations impacted training time.
>
> Moreover, in our TrivialAugment experiments (Sec.  6), the computational overhead of speckle reduction was negligible even with small models because each augmentation is applied less often. These findings show that these ultrasound-specific techniques can be included without substantial computational cost in practice. We have included these insights in Sec. 7.2 of the revised paper.
>
> **Applying TrivialAugment with only standard augmentations**
>
> Our evaluations of individual augmentations revealed that some standard augmentations ere ineffective or harmful for 13/14 tasks when applied in isolation (figures 8, 9). For this reason, we evaluated TrivialAugment using the top-$N$ most effective augmentations for each task (for different values of $N$), regardless of augmentation type.
>
> We can still determine whether adding ultrasound-specific augmentations is beneficial by examining tasks where the top-N augmentations are all standard, and observing whether adding the $(N+1)$-th US-specific augmentation improves performance. Across 10 tasks where the top-3 to top-8 augmentations are standard (Butterfly, CAMUS, GBCU and MMOTU classification; AUL liver, CAMUS, MMOTU, Open Kidney, PSFHS and Stanford Thyroid segmentation), adding an US-specific augmentation improved performance in only five cases. This supports our conclusion in Sec. 7.2 that researchers should evaluate whether standard augmentations suffice before investing resources into developing or using ultrasound-specific augmentations.

---

### Review · Reviewer_Kfqu · 2025-05-16

**Summary Of Contributions:**

This paper assesses a wide range of data augmentation techniques on ultrasound images, across a wide range of datasets, body parts, and problems in classification and segmentation. It also presents a benchmark for evaluating augmentation methods for ultrasound. Results are presented comparing the performance gains achieved by each individual augmentation and using a varying number of multiple augmentations.

**Audience:**

Yes

**Claims And Evidence:**

Yes

**Requested Changes:**

Below are my questions and comments. Please consider addressing them and changing the paper if helpful.

"we have created a benchmark of 14 different convex ultrasound image classification"
What does convex mean here?

In Figure 2, does each example stand for a 2D image, a 3D volume, or a subject?

In many of the datasets, one subject has multiple images. When performing classification task, did the authors predict the labels from a single image, or all the images from a subject? If the label is patient-level, e.g., normal/benign/malignant, then predicting from a single image may not be reasonable, as the pathology of interest may only exist in just a few slices among all images of a subject.

"For each task, we performed 30 independent repetitions of each augmentation."
Could the authors elaborate on this? What does "repetitions" mean here?

Could the authors elaborate a bit on "Brightness" augmentation? What does it do? Can it be replaced by normalization (i.e., normalizing the brightness of images to a common range)?

"In general, augmentation was less effective for segmentation than classification."
I'm not sure if it is valid to compare performance gains between classification and segmentation. They are different tasks, use different metrics, and has different baseline (no-augmentation) performances. Although the authors compare percentage change in metric, I'm not sure if the same percentage change indicates the same level of improvement in terms of e.g., impact on clinical practice.


"TrivialAugment transforms each image by randomly selecting two augmentations with replacement (including the possibility of no augmentation) from a predefined set, and applies them sequentially."
1. What's the different between TrivialAugment and a baseline multi-augmentation strategy? Is it that TrivialAugment selects two at a time and the baseline potentially uses all the augmentations on each image?
2. Did the authors apply TrivialAugment for all numbers of augmentations or just when using all 14 augmentations?


"..., we find that applying a diverse set of augmentations using TrivialAugment (Müller & Hutter, 2021) achieves substantial performance gains with limited tuning of the augmentation set."
Have the authors compared TrivialAugment with a baseline method for multiple augmentations?


"Evaluating each augmentation individually and applying TrivialAugment with the top three to five augmentations often maximized this trade-off and avoids detrimental augmentations. Adopting this approach is a good compromise between time consuming manual tuning and resource intensive automated optimization."
By "Evaluating each augmentation individually", did the authors mean training a model for each augmentation? What's the difference between this and "time consuming manual tuning"?

**Strengths And Weaknesses:**

Strengths:
1. The paper performs a well-organized and relatively comprehensive comparative study, presenting findings interesting to the field of ultrasound, and potentially the medical imaging community in general.
2. A good literature review on ultrasound-specific augmentation methods.
3. There are abundant details on the datasets, including the number of subjects, the dimension of images, etc.
4. Good findings about harmful augmentations, which seems less studied in literature.


Weaknesses:
1. The results and claims related to TrivialAugment could be better explained and supported.
2. The comparison between classification and segmentation results could be better justified.
3. Some details are missing or unclear.
Please see detailed comments below.

---

> ### Author Response · Authors · 2025-05-30
> **Responses to your feedback and questions**
>
> Reviewer Kfqu, thank you for your thoughtful and discerning feedback and recommendations. Please find our responses to your questions and descriptions of our changes based on your feedback below.
>
> **Image-level vs. patient-level evaluation**
>
> In all tasks, the models were trained and used to generate predictions for a single image. We agree that patient vs. image-level performance is an important distinction in many settings. Our choice to evaluate models based on per-image metrics (though still ensuring that there is no patient overlap between training and test splits) is due to constraints in the datasets and to standardize the evaluation protocol across the tasks. For example, patient-level predictions are not possible on the GBCU and MMOTU datasets because the patient IDs were removed by the original authors before publication. For the CAMUS, POCUS, and Butterfly classification tasks, although patient IDs are available the authors of the original studies also performed image-level evaluations. The one exception is the Fatty Liver dataset, where the original authors trained and evaluated their models using per-patient leave-one-out cross-validation. However, according to their description, the 10 frame sequence of images for each of the 55 patients was specifically selected from longer captured sequences. Therefore, we can have high-confidence that all images reflect the patient-level label (the presence or absence of nonalcoholic fatty liver disease) and are relevant for the task.
>
> **More detail on the “brightness” augmentation**
>
> The brightness adjustment augmentation increases or decreases the brightness of the image (using the RandomBrightnessContrast transform from Albumentations). It scales the pixel values of the image by a uniformly sampled value between an upper and lower bound (between 0.8 and 1.2 in our experiments). In contrast, it’s standard practice to normalize all images as a preprocessing step independent of data augmentation (although we uncovered a few cases where it has also been applied stochastically as an augmentation [1, 2], as shown in Fig. 2). In our experiments, we normalize the pixel values to the range [0, 1] as part of the preprocessing steps. We’ve updated Table 3 in Appendix B to include the class names of each augmentation (from the Albumentations library [4]) so that readers can more easily look up their definitions without having to turn to our source code to find them.
>
> **Comparing performance gains between classification and segmentation tasks**
>
> We have removed the comparisons between the magnitude of the improvements between segmentation and classification tasks when comparing the results between classification and segmentation tasks in Sec. 5.4. We agree that this should be interpreted with greater consideration of the setting in which the models would be used and we have more strongly emphasized this point when discussing the results in Sec. 7.1. Instead, in Sec. 5.4, we have narrowed the focus to the number and type of augmentations that are effective on the datasets for which there are both classification and segmentation tasks.
>
> **TrivialAugment and multiple augmentation experiments**
>
> The key distinction of TrivialAugment lies in its simplicity: it avoids tuning augmentation probabilities, strengths, or sequences. Despite this, it outperforms more complex automated methods—which themselves surpass manually designed strategies—on standard natural image benchmarks [3], and has seen broad adoption in other domains. We believe these qualities make it a strong baseline for demonstrating the possible performance gains using multiple augmentations.
>
> However, TrivialAugment relies on a predefined augmentation set. Our observation that certain augmentations were ineffective or harmful for specific tasks led us to evaluate TrivialAugment using augmentation sets comprising the top-$N$ augmentations per task, where $N \in \{2,… ,14\}$.
>
> When we suggest that using the top three to five augmentations with TrivialAugment offers a good trade-off, we refer to the relatively low cost of evaluating each augmentation individually (i.e., training a model per augmentation) compared to the extensive effort required to design and tune a strategy (i.e., augmentation probabilities, strengths and sequences) either manually or via automated search.
>
> Our goal is not to claim TrivialAugment is optimal, but to show that it offers a compelling balance between performance and effort for medical imaging domains—specifically ultrasound—when paired with a carefully selected augmentation set. We hope this encourages broader adoption of augmentation strategies in similar contexts.
>
> We have updated sections 6.1 and 7.3 to clarify our methodology and conclusions.
>
> *Please see the following comment for responses to your remaining questions.*

---

> > ### Author Response · Authors · 2025-05-30
> > **Responses to your feedback and questions (cont.)**
> >
> > **Other minor modifications and clarifications**
> >
> > - We have updated the introduction of Sec. 3 to clarify that the benchmark consists of tasks on “fan-shaped” ultrasound images captured with convex and phased array ultrasound probes (as opposed to linear probes, which generate rectangular images).
> > - We have updated the Y-axis labels of the subfigures in Fig. 2 to clarify that the counts are of the number of images for each class.
> > - “Repetitions” refers to repeated training runs with different random seeds (to account for randomness in the training process and in particular the application of data augmentations). We have made this clearer in the final paragraph of Sec. 5.1.
> >
> > **Additional References**
> >
> > [1] S. Basu, M. Gupta, P. Rana, P. Gupta, and C. Arora, “RadFormer: Transformers with global-local attention for interpretable and accurate Gallbladder Cancer detection,” MEDICAL IMAGE ANALYSIS, vol. 83, 2023.
> >
> > [2] S. Basu, M. Gupta, P. Rana, P. Gupta, and C. Arora, “Surpassing the Human Accuracy: Detecting Gallbladder Cancer from USG Images with Curriculum Learning,” in Proceedings of the IEEE/CVF Conference on Computer Vision and Pattern Recognition, 2022, pp. 20854–20864.
> >
> > [3] S. G. Müller and F. Hutter, “TrivialAugment: Tuning-Free Yet State-of-the-Art Data Augmentation,” in Proceedings of the IEEE/CVF International Conference on Computer Vision, 2021, pp. 774–782.
> >
> > [4] A. Buslaev, V. I. Iglovikov, E. Khvedchenya, A. Parinov, M. Druzhinin, and A. A. Kalinin, “Albumentations: Fast and Flexible Image Augmentations,” Information, vol. 11, no. 2, Art. no. 2, Feb. 2020, doi: 10.3390/info11020125.

---

> > > ### Comment · Reviewer_Kfqu · 2025-06-16
> > >
> > > Thank the authors for addressing my questions.

---

### Author Response · Authors · 2025-05-30
**Rebuttal Summary**

We thank the reviewers for their time reviewing our paper and for their insightful and positive feedback. We are encouraged that they find our analyses interesting (HpXM, Kfqu), rigorous (LQtm) and comprehensive (Kfqu); our work well motivated (LQtm); and our findings interesting for members of TMLR’s audience (HpXM, LQtm, Kfqu). We have responded to specific questions and requests for changes in individual responses to each review below.

**Key additions in our revised version:**
- Added results using larger models and transformers (App. E).
- Added a discussion of the limited computational cost of ultrasound-specific augmentations (Sec. 7.3, par. 3).
- Improved our comparisons of the effectiveness of augmentations between classification and segmentation tasks (Sec. 5.4, Sec. 7.1), acknowledging the caveats of such comparisons.
- Added clarification that our benchmark consists of tasks on “fan-shaped” ultrasound images captured with a convex and phased array ultrasound probes (Sec. 3, par. 1).
- Added a clarification that all model’s converged within the maximum number of epochs (Sec. 5.1, par. 3).

---

### Decision · Action_Editor_AJvc · 2025-06-27

**Recommendation:** Accept as is

**Additional Comments:**

This paper evaluates the impact of data augmentation on ultrasound (US) images. The authors propose using both general and ultrasound-specific augmentation techniques, which can be combined with TrivialAugment. Additionally, they introduce a benchmark for evaluating augmentation methods in the context of ultrasound imaging, addressing both classification and segmentation tasks.

The paper received initially positive feedback, with reviewers acknowledging its relevance to the community. However, they also highlighted several limitations, including the lack of diverse deep learning architectures in the experiments, as well as the need for clearer explanations of the evaluation protocol and the combination strategies for augmentations. The authors' rebuttal effectively addressed these concerns, and after the discussion period, the reviewers reached a consensus to accept the paper.

The AE has thoroughly reviewed the submission and the discussion. The AE concludes that the paper makes a valuable contribution to the state of the art, offering practical utility for research in medical image analysis with ultrasound. Therefore, the AE recommends acceptance.

**Audience:**

Yes

**Audience Explanation:**

The paper addresses the problem of data augmentation in ultrasound (US)  images — a topic of significant interest to the TMLR audience interested in medical applications.

**Claims And Evidence:**

Yes

**Claims Explanation:**

This is an interesting paper assessing the impact of augmentation in ultra sound (US) images. It provides valuable insights for both classification and segmentation tasks, while also introducing a novel evaluation protocol and benchmark datasets for performance assessment.